# UNPACKING SDXL TURBO: INTERPRETING TEXT-TO-IMAGE MODELS WITH SPARSE AUTOENCODERS

## ABSTRACT

Sparse autoencoders (SAEs) have become a core ingredient in the reverse engineering of large-language models (LLMs). For LLMs, they have been shown to decompose intermediate representations that often are not interpretable directly into sparse sums of interpretable features, facilitating better control and subsequent analysis. However, similar analyses and approaches were lacking for text-to-image models. We investigated the possibility of using SAEs to learn interpretable features for a few-step text-to-image diffusion models, such as SDXL Turbo. To this end, we train SAEs on the updates performed by transformer blocks within SDXL Turbo's denoising U-net. We find that their learned features are interpretable, causally influence the generation process, and reveal specialization among the blocks. In particular, we find one block mainly dealing with image composition, mainly responsible for adding local details, and, one for color, illumination, and style. Therefore, our work is an important first step towards better understanding the internals of generative text-to-image models like SDXL Turbo and showcases the potential of features learned by SAEs for the visual domain.

## 1 INTRODUCTION

Text-to-image generation is a rapidly evolving field. The DALL-E model first captured public interest (Ramesh et al., 2021), combining learned visual vocabularies with sequence modeling to produce high-quality images based on user input prompts. Today's best text-to-image models are largely based on text-conditioned diffusion models (Rombach et al., 2022; Saharia et al., 2022b; Podell et al., 2023; Sauer et al., 2023b; Betker et al., 2023; Pernias et al., 2023). This can be partially attributed to diffusion models' stable training dynamics, which makes them easier to scale than previous approaches like generative adversarial neural networks (Dhariwal & Nichol, 2021). As a result, they can be trained on internet scale image-text datasets (Schuhmann et al., 2022a) and learn to generate photorealistic images from text.

However, the underlying logic of the neural networks enabling the text-to-image pipelines we have today, due to their black box nature, is not well understood. Unfortunately, this lack of interpretability is typical in the deep learning field. For example, advances in image recognition (Krizhevsky et al., 2012) and language modeling (Devlin, 2018; Brown, 2020) come mainly from scaling models (Hoffmann et al., 2022), rather than from an improved understanding of their internals. Recently, the emerging field of mechanistic interpretability has sought to alleviate this limitation by reverse engineering visual models (Olah et al., 2020) and transformer-based LLMs (Rai et al., 2024). At the same time, diffusion models have remained under-explored.

This work focuses on SDXL Turbo, a recent open-source few-step text-to-image diffusion model. We leverage methodologies originally developed for language models, which allow inspection of the intermediate results of the forward pass (Chen et al., 2024; Ghandeharioun et al., 2024; Cunningham et al., 2023; Bricken et al., 2023). Moreover, some even enable reverse engineering the entire task-specific subnets (Marks et al., 2024). In particular, *sparse autoencoders (SAEs)* (Yun et al., 2021; Cunningham et al., 2023; Bricken et al., 2023) are considered a breakthrough in interpretability for LLMs. They have been shown to decompose intermediate representations of the LLM forward pass – often difficult to interpret due to *polysemanticity*[1] – into sparse sums of interpretable and

---

[1]A phenomenon where a single neuron or feature encodes multiple, unrelated concepts (Elhage et al., 2022)

monosemantic features. These features are learned in an unsupervised way, can be automatically annotated using LLMs (Caden et al., 2024), and facilitate subsequent analysis, for example, circuit extraction (Marks et al., 2024).

**Contributions.** In this work, we ask whether we can use SAEs to draw information on the computation performed by the 1-step generation process of SDXL Turbo, a recent open-source few-step text-to-image diffusion model.

To facilitate our analysis, we developed a library called *SDLens* that allows us to cache and manipulate intermediate results of SDXL Turbo's forward pass. We use our library to create a dataset of SDXL Turbo's intermediate feature maps of several transformer blocks inside SDXL Turbo's U-net on 1.5M LAION-COCO prompts (Schuhmann et al., 2022a;b). We then use these feature maps to train multiple SAEs for each transformer block.[2] Finally, we perform a quantitative and qualitative analysis of the SAE's learned feature maps:

1. We empirically show the potential of SAEs to learn highly interpretable features in diffusion-based text-to-image models.
2. We developed visualization techniques to analyze the interpretability and causal effects of the learned features.
3. We perform two case studies in which we visualize and interpret the active features in different transformer blocks, finding evidence that certain transformer blocks of SDXL Turbo's pipeline specialize in *image composition*, *adding details*, and *style*.
4. We follow up our qualitative case studies by designing multiple quantitative experiments that show that our hypotheses hold up also on larger sample sizes.
5. As a part of our quantitative analysis, we create an automatic feature annotation pipeline for the transformer block which appears responsible for *image compositions*.

**Note on Visualizations.** Qualitative analysis through inspection of generated images is crucial for this type of research. However, since image grids require a significant amount of space, many essential visualizations are included into App. F and the supplementary material.

## 2 BACKGROUND

### 2.1 SPARSE AUTOENCODERS

Let $h(x) \in \mathbb{R}^d$ be some intermediate result of a forward pass of a neural network on the input $x$. In a fully connected neural network, the components $h(x)$ could correspond to neurons. In transformers, which are residual neural networks with attention and fully connected layers, $h(x)$ usually either refers to the content of the residual stream after some layer, an update to the residual stream by some layer, or the neurons within a fully connected block. In general, $h(x)$ could refer to anything, e.g., also keys, queries, and values. It has been shown (Yun et al., 2021; Cunningham et al., 2023; Bricken et al., 2023) that in many neural networks, especially LLMs, intermediate representations can be well approximated by sparse sums of $n_f \in \mathbb{N}$ learned feature vectors, i.e.,

$$h(x) \approx \sum_{\rho=1}^{n_f} s_\rho(x) \mathbf{f}_\rho, \tag{1}$$

where $s_\rho(x)$ are the input-dependent[3] coefficients most of which are equal to zero and $\mathbf{f}_1, \ldots, \mathbf{f}_{n_f} \in \mathbb{R}^d$ is a learned dictionary of feature vectors. Importantly, the features are usually interpretable.

**Sparse autoencoders.** In order to implement the sparse decomposition from equation 1, the vector $s$ containing the $n_f$ coefficients of the sparse sum is parameterized by a single linear layer followed by ReLU activations, called the *encoder*,

$$s = \text{ENC}(h) = \sigma(W^{\text{ENC}}(h - b_{\text{pre}}) + b_{\text{act}}), \tag{2}$$

in which $h \in \mathbb{R}^d$ is the latent that we aim to decompose, $\sigma(\cdot) = \max(0, \cdot)$, $W^{\text{ENC}} \in \mathbb{R}^{n_f \times d}$ is a learnable weight matrix and $b_{\text{pre}}$ and $b_{\text{act}}$ are learnable bias terms. We omitted the dependencies $h = h(x)$ and $s = s(h)$ that are clear from context.

---

[2] We plan to release our library and trained SAEs to allow other scholars and the OSS community to replicate and extend our findings conveniently.

[3] In the literature this input dependence is usually omitted.

Similarly, the learnable features are parametrized by a single linear layer, called *decoder*,

$$h' = \text{DEC}(s) = W^{\text{DEC}}s + b_{\text{pre}}, \qquad (3)$$

in which $W^{\text{DEC}} = (\mathbf{f}_1 | \cdots | \mathbf{f}_{n_f}) \in \mathbb{R}^{d \times n_f}$ is a learnable matrix of whose columns take the role of learnable features and $b_{\text{pre}}$ is a learnable bias term.

**Training Details.** An extended version of this section including training details can be found in App. A.

## 2.2 FEW STEP DIFFUSION MODELS: SDXL TURBO

**Diffusion models.** Diffusion models (Sohl-Dickstein et al., 2015; Ramesh et al., 2022; Rombach et al., 2022; Saharia et al., 2022a) sample from an unknown distribution $p$ by learning to iteratively denoise corrupted samples, starting from pure noise. The corruption process is defined on training samples from $p$. Mathematically, the images are corrupted with Gaussian noise and are distributed according to

$$q_t(x_t|x_0) := \mathcal{N}(\alpha_t x_0, \sigma_t^2 \mathbf{I}), \qquad (4)$$

where $x_0$ corresponds to a real image from $p$, $0 \leq t \leq T$, $\alpha_t, \sigma_t^2$ are positive real-valued scalars such that the signal-to-noise ratio $SNR := \frac{\alpha_t}{\sigma_t^2}$ is monotonically decreasing. Additionally, The coefficients $\alpha_{T-1}, \sigma_{T-1}^2$ are typically chosen such that $x_T \sim \mathcal{N}(0, \mathbf{I})$.

The denoising process is implemented via a learned distribution $p_\theta(x_{t-1}|x_t)$. The simplest way to generate samples using $p_\theta(x_{t-1}|x_t)$ is to first generate a sample of pure noise $x_T \sim \mathcal{N}(0, \mathbf{I})$, followed by $T$ iterative applications of $p_\theta$, which yields a sequence $x_T, x_{T-1}, ..., x_1, x_0$, where $x_0$ approximates samples from $p$. The vector $\theta$ represents the parameters of a neural network that defines $p_\theta(x_{t-1}|x_t)$. The denoising distribution $p_\theta(x_{t-1}|x_t)$ is parameterized to be Gaussian.

The neural network used to parameterize $p_\theta(x_{t-1}|x_t)$ can be trained to learn different quantities (Luo, 2022; Salimans & Ho, 2022; Karras et al., 2022). A possible approach is to directly output the mean $\mu_t$ of $p_\theta(x_{t-1}|x_t)$, while the variance is either fixed or learned as well. In this work, the neural network is parameterized to predict the noise added to the original sample during the forward process (eq. (18)). This is achieved by minimizing the objective $w(t)\|\epsilon - \epsilon_\theta(\alpha_t x_0 + \sigma_t \epsilon, t)\|^2$, where $w$ is a weighing function (Ho et al., 2020). Once $\epsilon_\theta$ is trained, the mean of $p_\theta(x_{t-1}|x_t)$ is computed as $\frac{1}{\alpha_t}(x_t - \sigma_t \epsilon_\theta)$ (Rombach et al., 2022). Since our primary goal is to analyze a pre-trained diffusion model, we refer the interested reader to App. B.

**Latent diffusion.** Originally, diffusion models operated directly on pixels (Ho et al., 2020; Song & Ermon, 2020). However, training a denoising network in pixel space is computationally expensive (Hoogeboom et al., 2023). Thus, Rombach et al. (2022) uses a pre-trained autoencoder to first compress images and define a diffusion process in the latent space instead. To make this difference clear they write $p_\theta(z_{t-1}|z_t)$, in which now $z_t$ refers to a noisy latent instead of a noisy image.

**SDXL Turbo.** To speed-up inference of latent diffusion models, Sauer et al. (2023b) distills a pre-trained Stable Diffusion XL (SDXL) model (Podell et al., 2023). The distilled model is referred to as *SDXL Turbo* as it allows high-quality sampling in as little as 1-4 steps. In comparison, the original original SDXL model is trained with a noise schedule of 1000 steps, but in practice, sampling with 20 to 50 steps still generates high-quality images.

**Neural network architecture.** The denoising network of *SDXL Turbo* estimating $p_\theta(z_{t-1}|z_t)$ is implemented using a U-net similar to Rombach et al. (2022). The U-net is composed of a down-sampling path, a bottleneck, and an up-sampling path. Both the down-sampling and up-sampling paths are composed of 3 individual blocks. The individual block structure differs slightly but both down- and up-sampling blocks consist of residual layers with some blocks including cross-attention transformer layers while others do not. Finally, the bottleneck layer is also composed of attention and residual layers. Importantly, the text conditioning is achieved via cross-attention to text embeddings performed by in total 11 transformer blocks embedded in the down-, up-sampling path and bottleneck. An architecture diagram displaying the relevant blocks can be found in App. B Fig. 2.

## 3 SPARSE AUTOENCODERS FOR SDXL TURBO

With the necessary definitions at hand, in this section we show a way to apply SAEs to SDXL Turbo. In the following, we assume that all SDXL Turbo generations are done using a 1-step process.

**Where to apply the SAEs.** We apply SAEs to the updates performed within the cross-attention transformer blocks responsible for incorporating the text prompt (depicted in Fig. 2). Each of these blocks consists of multiple transformer layers, which attend to all spatial locations (self-attention) and to the text prompt embeddings (cross-attention). Since the overall architecture is a U-net, the shapes feature maps can get manipulated inbetween the cross-attention blocks by residual network blocks, upscaling layers and downscaling layers.

Formally, the cross-attention transformer blocks update their inputs in the following way

$$D_{ij}^{out} = D_{ij}^{in} + \text{TRANSFORMER}(D^{in}, c)_{ij}, \tag{5}$$

in which $D^{in}, D^{out} \in \mathbb{R}^{h \times w \times d}$ denote the residual stream before and after application the cross-attention block respectively. The transformer block itself calculates the function $\text{TRANSFORMER}[\ell] : \mathbb{R}^{h \times w \times d} \to \mathbb{R}^{h \times w \times d}$. Note that we denote the input to the cross-attention transformer block as $D^{in}$ to highlight that there can be layers in between cross-attention transformer block $\text{TRANSFORMER}[\ell]$ and the previous one $\text{TRANSFORMER}[\ell-1]$. Further, we omitted the input noise $z_t$ and text embedding $c$ and the block index $\ell$ for both $D[\ell]^{in}(z_t, c)$ and $D[\ell]^{out}(z_t, c)$.

We train our SAEs on the residual updates $\text{TRANSFORMER}[\ell](D^{in}, c)_{ij} \in \mathbb{R}^d$, we denote them by

$$\Delta D_{ij} := \text{TRANSFORMER}(D^{in}, c)_{ij} = D_{ij}^{out} - D_{ij}^{in}. \tag{6}$$

That is, we train one encoder $\text{ENC}[\ell]$, decoder $\text{DEC}[\ell]$ pair per transformer block $\ell$ and share it over all spatial locations $i, j$. For notational convenience we omit block indices from now. We do this the for 4 out of the 11 transformer blocks (App. B Fig. 2) that we found have the highest impact on the generation (see App. C), namely, down.2.1, mid.0, up.0.0 and up.0.1.

**Feature maps.** We refer to $\Delta D \in \mathbb{R}^{h \times w \times d}$ as dense feature map and applying $\text{ENC}$ to all image locations results in the *sparse feature map* $S \in \mathbb{R}^{h \times w \times n_f}$ with entries

$$S_{ij} = \text{ENC}(\Delta D_{ij}). \tag{7}$$

We refer to the feature map of the $\rho$th learned feature using $S^\rho \in \mathbb{R}^{h \times w}$. This feature map $S^\rho$ contains the spatial activations (or coefficients) of the $\rho$th learned feature $\mathbf{f}_\rho \in \mathbb{R}^d$, which is a column in the decoder matrix $W^{\text{DEC}} = (\mathbf{f}_1 | \cdots | \mathbf{f}_{n_f}) \in \mathbb{R}^{d \times n_f}$. Therefore, now we can represent each element of the dense feature map as a sparse sum

$$\Delta D_{ij} \approx \sum_{\rho=1}^{n_f} S_{ij}^\rho \mathbf{f}_\rho, \text{ with } S_{ij}^\rho = 0 \text{ for most } \rho \in \{1, \dots, n_f\}. \tag{8}$$

**Training.** In order to train an SAE for a transformer block, we collected dense feature maps $\Delta D_{ij}$ from SDXL Turbo one-step generations on 1.5M prompts from the LAION-COCO (Schuhmann et al., 2022b). Each feature map has dimensions of $16 \times 16$, resulting in a training dataset of 384M dense feature vectors per transformer block. For the SAE training process, we followed the methodology outlined in (Gao et al., 2024), using the TopK activation function and an auxiliary loss to handle dead features. For more details see App. A.

## 4 QUALITATIVE ANALYSIS OF THE TRANSFORMER BLOCKS

Here we introduce feature visualization techniques and use them to qualitatively analyse the learned features. While we have one SAE per transformer block, we omit the transformer block index $\ell$.

### 4.1 FEATURE VISUALIZATION TECHNIQUES

We start by introducing feature visualization techniques and use them to depict the features active for the prompt "A cinematic shot of a professor sloth wearing a tuxedo at a BBQ party." (see Fig. 5).

**Spatial activations.** We visualize the feature map $S^\rho \in \mathbb{R}^{h \times w}$ containing the activations of a feature $\rho$ across the spatial locations by upscaling it to the size of the generated images and overlaying it as a heatmap over the generated images. In the heatmap red encodes the highest feature activation and blue the lowest nonzero activation. For examples, have a look at the first column of Fig. 5.

**Top dataset examples.** For a given feature $\rho$ we can sort our dataset examples according to their average spatial activation

$$a_\rho = \frac{1}{wh} \sum_{i=1}^{h} \sum_{j=1}^{w} S_{ij}^\rho \in \mathbb{R}. \tag{9}$$

We use equation 9 to define top dataset examples and to sample from the top 5% quantile of activating examples ($a_\rho > 0$). Further, we will refer to them as top 5% images for a feature $\rho$.

Note that $S_{ij}^\rho$ always depends on an input prompt embedding $c$ and input noise $z_1$, via $S_{ij}(c, z_1) = \text{ENC}(\Delta D_{ij}(c, z_1))$, which we usually omit for ease of notation. As a result $a_\rho$ also depends on $c$ and $z_1$. When we refer to top dataset examples we mean our $(c, z_1)$ pairs ones with the largest values for $a_\rho(c, z_1)$. For examples, have a look at the last four columns of Fig. 5.

**Activation modulation.** We design interventions that allow us to modulate the strength of the $\rho$th feature. We do so, by adding or subtracting a multiple of the feature $\rho$ on all of the spatial locations $i, j$ proportional to its original activation $S_{ij}^\rho$

$$\Delta D_{ij}' = \Delta D_{ij} + \beta S_{ij}^\rho \mathbf{f}_\rho, \tag{10}$$

in which $\Delta D_{ij}, \Delta D_{ij}'$ is the update performed by the transformer block before and after the intervention, $\beta \in \mathbb{R}$ is a modulation factor, and $\mathbf{f}_\rho$ is the $\rho$th learned feature vector. For examples, have a look at the columns with titles containing "A" in Fig. 5. The titles contain $\beta$'s value respectively.

We observed that often positive $\beta$ values significantly greater than one are required to causally affect the output. Similarly, while turning off a feature by setting it to zero, which is equivalent to $\beta = -1$, we observed that significantly smaller negative values are required for feature ablations to causally affect the output. Interestingly, while such coefficients never occur naturally by forward passing noise and text embeddings their effects are usually interpretable. This is akin to what is observed when applying steering techniques (Rimsky et al., 2023; Geiger et al., 2024) or SAE interventions to language models (Cunningham et al., 2023; Bricken et al., 2023).

**Activation on empty context.** Another way of visualizing the causal effect of features is to activate them while doing a forward pass on the empty prompt $c(\text{""})$. In order to do so, we turn off all other features at the transformer block $\ell$ of intervention and turn on the target feature $\rho$. Formally, modify the forward pass by setting

$$D_{ij}^{out'} = D_{ij}^{in} + \gamma k \mu_\rho \mathbf{f}_\rho, \tag{11}$$

in which $D_{ij}^{out'}$ replaces residual stream plus transformer block update, $D_{ij}^{in}$ is the input to the block, $\mathbf{f}_\rho$ is the $\rho$th learned feature vector, $\gamma \in \mathbb{R}$ is a hyperparameter to adjust the intervention strength, and $\mu_\rho$ is a feature dependent multiplier obtained by taking the average activation across positive activations of $\rho$ (collected over a subset of 50.000 dataset examples). Multiplying it by $k$ aims to recover the coefficients lost by setting the other features to zero. For examples, have a look at the columns with titles containing "B" in Fig. 5. Again $\gamma$'s value is in the titles.

Note that for both intervention types, we directly added/subtracted feature vectors to the dense vectors, instead of encoding, manipulating sparse features, and decoding. By doing so, we mitigate side-effects caused due to reconstruction loss.

### 4.2 CASE STUDY I: MOST ACTIVE FEATURES ON A PROMPT

Combining all our feature visualization techniques, in Fig. 1, we depict the features with the highest average activation when processing the prompt: "A cinematic shot of a professor sloth wearing a tuxedo at a BBQ party". We discuss the transformer blocks in order of decreasing interpretability. In App. E Fig. 5 we have the same case study but with top 9 instead of top 5).

**Down.2.1.** The `down.2.1` transformer block indeed seems to contribute to the image composition. Several features relate to the prompt: 4539 "professor sloth", 4751 "a tuxedo", 2881 "party".

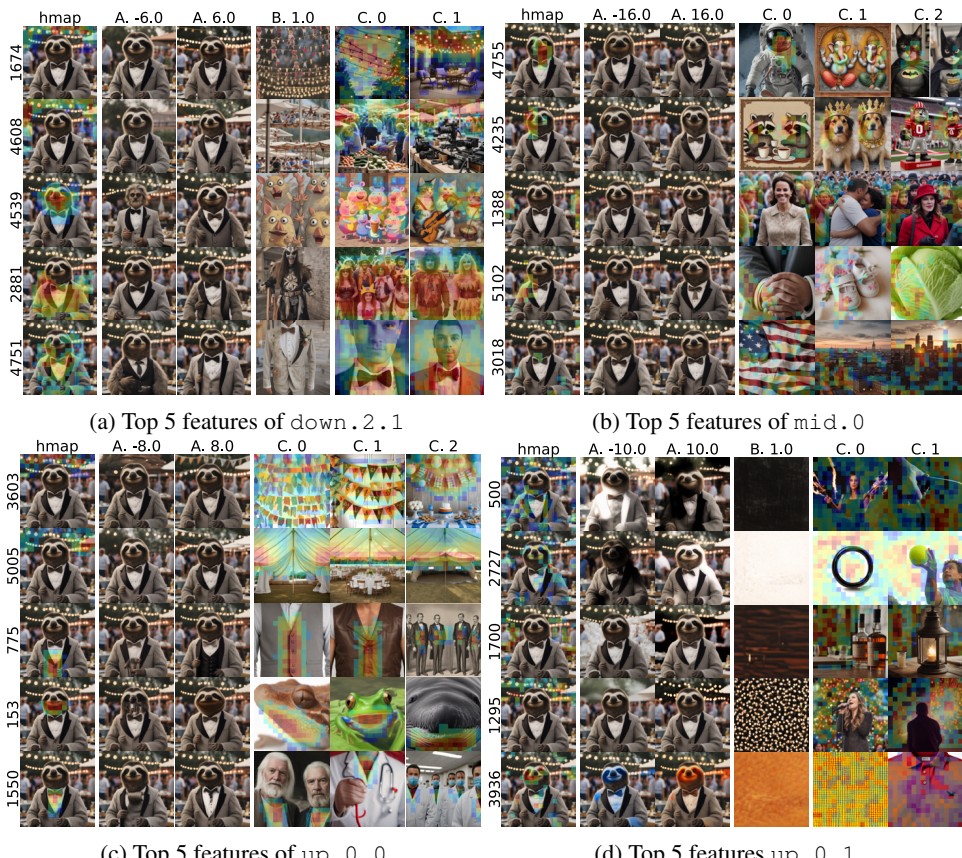

Figure 1: The top 5 features of `down.2.1` (a), `mid.0` (b), `up.0.0` (c) and `up.0.1` (d) for the prompt: "A cinematic shot of a professor sloth wearing a tuxedo at a BBQ party." Each row represents a feature. The first column depicts a feature heatmap (highest activation red and lowest nonzero one blue). The column titles containing "A" show feature modulation interventions, the ones containing "B" the intervention of turning on the feature on the empty prompt, and the ones containing "C" depict top dataset examples. Floating point values in the title denote $\beta$ and $\gamma$ values.

Turning off features (A. -6.0 column) removes elements from and changes elements in the scene in ways that mostly make sense when comparing with the heatmap (hmap column) and the top examples (C columns): 1674 *removes* the light chains in the back, 4608 the umbrellas/tents, 4539 the 3D animation-like sloth face, and, 4751 *changes* the type of suit. Similarly, enhancing the same features (A. 6.0 column) enhances the corresponding elements and sometimes changes them.

Activating the features on the empty prompt often creates related elements. With the fixed random seed we use, the empty prompt itself looks like a painting of a piece of nature with a lot of green and brown. Therefore, while the prompt is empty, the features active during the forward pass that are not and due to the transformer blocks that we don't intervene on still contribute to the images.

While top dataset examples (C.0, C.1) and empty prompt intervention (B.) mostly agree with the feature activation heatmaps (hmap column), some of them provide additional insights, e.g., 2881 which activates on the suit, seems to correspond to (masqueraded) characters in a (festive) scene.

**Up.0.1.** The features of `up.0.1` indeed seem to contribute to the style. They only indirectly relate to the prompt. The illumination (2727) and shadow (500, 1700) effects relate to "a cinematic shot".

Interestingly, turning on the `up.0.1` features on the entire empty prompt (B. column) results in texture-like images. In contrast, when activating them locally (A. columns) their contribution to the output is highly localized and keeps most of the remaining image largely unchanged. For the `up.0.1` we find it remarkable that often the ablation and amplification are counterparts: 500 (light, shadow), 2727 (shadow, light), 3936 (blue, orange).

**Up.0.0.** For the third, `up.0.0`, we observe that it acts locally and seems to require relevant other features from the other transformer blocks to effectively influence the image. For the empty prompt, activating these features results in abstract looking images, which are hard to relate to the other columns. Thus, we excluded this visualization technique and instead added one more example.

Most top dataset examples and their activations (C columns) are highly interpretable: 3603 party decoration, 5005 upper part of tent, 775 buttons on suit, 153 lower animal jaw, 1550 collars. All of the features have an expected causal effect on the generation when ablating/enhancing (B. columns): 3603, 5005, 775, 153, 1550. To sum up, the learned features of this transformer block primarily adds local details to the generation. The interventions are effective only given a suitable context.

**Mid.0.** The specific role of the forth, `mid.0` is not well understood. We find it harder to interpret because most interventions on the `mid.0` have very subtle effects. We did not include the empty prompt intervention because they barely affect the generation.

While effects of interventions are subtle, dataset examples (C. columns) and heatmap (hmap column) all mostly agree with each other and are specific enough to be interpretable: 4755 bottom right part of faces, 4235 left part of (animal) faces, 1388 people in the background, and, 5102 outlines the left border of the main object in the scene. We hypothesize that `mid.0`'s features are more abstract, indicate where things are[4] and potentially how they relate to each other.

### 4.3 Case study II: Random features

Next, we look at the learned features in isolation, i.e., independently of the context of a specific prompt. In App. F Fig. 6, Fig. 7, and Fig. 8,[5] we do so by selecting dataset examples that have the largest average activation and perturbing them using our activation modulation intervention from above. For `down.2.1` and `up.0.1` we also include the empty prompt interventions.

**Interpretation.** Overall, on the random feature examples, we can make similar observations as in Fig. 5 where we looked at the most active features when generating a picture for a prompt. Namely,

1. `down.2.1` mostly creates elements in the scene of which many directly correspond to phrases in the prompt. Both its global as well as local interventions are highly effective and interpretable. To term it a *compositional block* is not too far off.
2. `mid.0` features seem to be abstract, contain information about where things are (0 bottom left corner, 2 sides of a woman/person, 4 top right object boundary in App. F Fig. 6) and maybe also how they relate to each other (3 object inside other object, 5 bottom right of a kind). Global interventions on the empty prompt barely change the generation and local interventions result in subtle changes.
3. `up.0.0` features mostly correspond to concrete details in the images. While empty-prompt interventions result in abstract-looking images, local interventions in context are highly effective and change images in expected ways. We term `up.0.0` a *detail block*.
4. `up.0.1` features mostly correspond to stylistic aspects of the image such as colour, texture, illumination and shadow. Both global as well as local interventions are highly effective and interpretable. We find the folklore term of *style block* appropriate.

When studying features in isolation, it becomes apparent that distinctions between the blocks are not clear cut. While `down.2.1` has the most features related to the prompt, all transformer blocks usually have some features that directly relate to phrases in the prompt. E.g., 1 in `mid.0`: "ceiling fan"[6]. Some `down.2.1` features are style features as well, e.g., it usually has a *anime style* and a *cartoon style* feature (see App. F Fig. 8). The difference inbetween `down.2.1` style features and `up.0.1` ones is maybe that the former ones are acting more globally, while the latter ones usually only change individual aspects (App. F Fig. 10). Some `up.0.0` features also directly relate to phrases in the prompt and some of them also have an effect when turning them on in the empty

---

[4]SDXL Turbo does not utilize positional encodings for the spatial locations in the feature maps. Therefore, we did a brief sanity check and trained linear probes to detect $i, j$ given $D_{ij}^{in}$. These probes achieved high accuracy on a holdout set: $97.9\%, 98.48\%, 99.44\%, 95.57\%$ for `down.2.1, mid.0, up.0.0, up.0.1`.

[5]For the features in App. F Fig. 8 we provide additional visualizations in which we turn the corresponding features on on unrelated prompts (Fig. 9) and the `up.0.1` ones locally in Fig. 10.

[6]For lack of space, we included the prompts corresponding to all features in Fig. 6 only in App. F Table 3.

prompt. Some `up.0.1` features don't only change style but also add and remove elements of the scene, e.g., dog eyes (Fig. 6 `up.0.1` feature 3).

## 5 QUANTITATIVE INTERPRETING THE LEARNED FEATURES

### 5.1 ANNOTATION PIPELINE

Feature annotation with an LLM followed by further evaluation is a common way to assess feature properties such as interpretability, specificity, and causality of learned features (Caden et al., 2024). We found it applicable to the `down.2.1` transformer block learned features, which have a strong effect on the generation and thus are amendable to automatic annotation using visual language models such as GPT-4o (OpenAI, 2024). In contrast, for features of other blocks with more subtle effects, we found VLM-generated captions to be unsatisfactory. In order to caption the features of `down.2.1`, we prompt GPT-4o with a sequence of 14 images. The first five images are irrelevant to the feature (i.e., the feature was inactive during the generation of the images), followed by a progression of 4 images with increasing average activation values, and finished five images with the highest average activation value. The last nine images are provided alongside their so-called "coldmaps": a version of an image with weakly active and inactive regions being faded and concealed. The prompt template and examples of the captions can be found in App. G.

### 5.2 EXPERIMENTAL DETAILS

We perform a series of experiments in order to get statistical insights into the features learned. We will report the majority of experimental score in the format $M(S)$. When the score is reported in the context of SDXL Turbo's transformer block, it means that we computed the score for each feature of the block and set $M$ and $S$ to mean and standard deviation across the feature scores. For the baselines, we calculate the mean and standard deviation across the scores of a 100-element sample.

Table 1: Metrics for SDXL Turbo blocks and baselines.

(a) Specificity, texture score, and color activation for different blocks and baselines.

| Block | Specificity | Texture | Color |
|---|---|---|---|
| Down.2.1 | 0.71 (0.11) | 0.16 (0.02) | 86.2 (14.9) |
| Mid | 0.62 (0.11) | 0.14 (0.01) | 84.7 (16.3) |
| Up.0.0 | 0.66 (0.12) | 0.18 (0.03) | 86.3 (16.5) |
| Up.0.1 | 0.65 (0.11) | 0.20 (0.02) | 73.8 (20.6) |
| Random | 0.50 (0.10) | 0.13 (0.02) | 90.7 (54.9) |
| Same Prompt | 0.89 (0.06) | – | – |
| Textures | – | 0.18 (0.02) | – |

(b) Manhattan distances between original and intervened images at varying intervention strengths.

| Block | -10 | -5 | 5 | 10 |
|---|---|---|---|---|
| Down.2.1 | 148.2 / 116.0 | 124.2 / 94.4 | 101.4 / 78.7 | 128.9 / 105.60 |
| Mid | 69.2 / 32.2 | 39.4 / 18.5 | 33.2 / 15.2 | 59.9 / 29.82 |
| Up.0.0 | 105.3 / 38.4 | 77.7 / 23.7 | 63.6 / 23.3 | 88.6 / 37.08 |
| Up.0.1 | 125.0 / 26.8 | 73.1 / 16.4 | 68.6 / 21.9 | 98.9 / 34.74 |

**Interpretability.** Features are usually considered interpretable if they are sufficiently specific, i.e., images exhibiting the feature share some commonality. In order to measure this property, we compute the similarity between images on which the feature is active. High similarity in between these images is a proxy for high specificity. For each feature, we collect 10 random images among top 5% images for this feature and calculate their average pairwise CLIP similarity (Radford et al., 2021; Cherti et al., 2023). This value reflects how semantically similar the contexts are in which the feature is most active. We display our results in the first column of Table 1 (a), which shows that the CLIP similarity between images with the feature active is significantly higher then the random baseline for all transformer blocks. This suggests that when a feature is on, the images are similar.

For `down.2.1` we compute an additional *interpretability* score by comparing how well the generated annotations align with the top 5% images. The resulting CLIP similarity score is 0.21 (0.03) and again significantly higher then the random baseline (average CLIP similarity with random images) 0.12 (0.02). To obtain an upper bound on this score we also compute the CLIP similarity to an image generated from the feature annotation, which is 0.25 (0.03).

**Causality.** We can use the feature annotations to measure a feature's causal strength by comparing the empty prompt intervention images with the caption.[7] The CLIP similarity in between inter-

---

[7] Since we require captions for our quantitative causality analysis we only have it for `down.2.1`.

vention images and feature caption is 0.19 (0.04) and almost matches the annotation based interpretability score of 0.21 (0.03). This suggests that empty prompt intervention images are similar to the corresponding feature annotations even though the annotation pipeline has never seen such images, which speaks for the high causal strength of features learned on `down.2.1`.

**Sensitivity.** A feature is said to be sensitive when it activates on its relevant context. As a proxy for the context, we have chosen the feature annotations obtained with the auto-annotation pipeline.[8] For each learned feature, we collected the 100 prompts from a 1.5M sample of LAION-COCO with the highest sentence similarity based on sentence transformer embeddings of `all-MiniLM-L6-v2` Reimers & Gurevych (2019). We make sure that the resulting set of 100 prompts is diverse. Next, we run SDXL Turbo on these prompts and count the proportion of generated images in which the feature is active on more than 0%, 10%, 30% of the image area, resulting in 0.60 (0.32), 0.40 (0.34), 0.27 (0.30) respectively, which is much higher than the random baseline, which is at 0.06 (0.09), 0.003 (0.006), 0.001 (0.003). However, the average scores are $< 1$ and thus not perfect. This may be caused by incorrect or imprecise annotations for the features that are subtle and, therefore, hard to annotate with a VLM and SDXL Turbo failing to comply with some prompts.

**Relatedness to texture.** In Fig. 5 and App. F Fig. 6 the empty prompt interventions of the `up.0.1` features resulted in texture-like pictures. In order to quantify whether this consistently happens, we design a simple texture score by computing the CLIP similarity between an image and the word "texture". Using this score, we compare empty prompt interventions of the different transformer blocks with each other and also to real-world texture images. The results are in the second column of Table 1 (a) and suggest that `up.0.1` and `up.0.0` generate textures and some of the `down.2.1` images look like textures as well. For `up.0.0` we did not observe any connection of these images to the top activating images. In particular, the score of `up.0.1` is higher than the one of the real-world textures dataset (Cimpoi et al. (2014)).

**Color sensitivity.** In our qualitative analysis, we suggested that the features learned on `up.0.1` relate to texture and color. If this holds, the image regions that activate a feature should not differ significantly in color on average. To test that, we calculate the "average" color for each feature: this is a weighted average of pixel colors with the weights as activation values among random 10 of top 5% images for this feature. Then, we calculate the weighted average Manhattan distance between the colors of the pixels and the "average" color on the same images (the highest possible distance is $3 \cdot 255 = 765$). We report these distances for different transformer blocks and for the images generated on random prompts from LAION-COCO. We present our results in the third column of Table 1 (a). The average distance for the `up.0.1` transformer block is in fact the lowest.

**Intervention locality.** We suggested that the features learned on `up.0.0` and `up.0.1` influence intervened generations locally. To quantitatively assess that, we estimate how the top 5% images change inside and outside the active regions. In order to exclude weak activation regions from consideration, we say that a pixel is inside the active area if the corresponding 32x32 patch has an activation value larger than 50% of the image patches, and it is outside the active area, if the corresponding 32x32 patch has activation value of zero. In the Table 1 (b), we report Manhatten distances between the original images and the intervened images inside and outside the active areas for intervention strengths -10, -5, 5, 10. The features for `up.0.0` and `up.0.1` have a higher effect inside the active area than outside, in contrast to `down.2.1` for which this difference is smaller.

## 6 RELATED WORK

**Analyzing the latent space of diffusion models.** Kwon et al. (2023) show that diffusion models naturally have a semantically meaningful latent space. Park et al. (2023) analyzes the latent space of DMs using Riemannian geometry. Li et al. (2024); Dalva & Yanardag (2024) presents self-supervised methods for finding semantic directions in the latent space. Similarly, Gandikota et al. (2023) show that the attribute variations lie in a low-rank space by learning LoRA adapters (Hu et al., 2021) on top of pre-trained DMs. Brack et al. (2023); Wang et al. (2023) demonstrate effective semantic vector algebraic operations in the latent space of DMs, as observed by Mikolov et al. (2013). However, none of those works train SAEs to interpret and control the latent space explicitly.

---

[8]Since we require feature annotations for our sensitivity analysis, we only have it for `down.2.1`.

**Mechanistic interpretability using SAEs.** Sparse autoencoders have only recently been popularized by Bricken et al. (2023), in which they show that it is possible to learn interpretable features by decomposing neurons in MLPs in 2-layer transformer language models. At the same time, a parallel work decomposed the elements of the residual stream (Cunningham et al., 2023), which followed up on (Sharkey et al., 2022). To our knowledge, the first work that applied sparse autoencoders to transformer-based LLM was (Yun et al., 2021), which learned a joint dictionary for patterns of all layers. Recently, sparse autoencoders have gained a lot of traction and many have been trained even on state-of-the-art LLMs (Gao et al., 2024; Templeton & et al., 2024; Lieberum et al., 2024). In addition, great tools are available for inspection (Lin & Bloom, 2023) and automatic interpretation Caden et al. (2024) of learned characteristics. Marks et al. (2024) have shown how to use SAE features to facilitate automatic circuit discovery.

The works most closely related to our work are (Ismail et al., 2023) and (Daujotas, 2024). Ismail et al. (2023) apply concept bottleneck methods (Koh et al., 2020) that decompose latent concepts into vectors of interpretable concepts to generative image models, including diffusion models. Unlike the SAEs that we train, this method requires labeled concept data. Daujotas (2024) decomposes CLIP (Radford et al., 2021; Cherti et al., 2023) vision embeddings using SAEs and use them for conditional image generation with a diffusion model called Kandinsky (Razzhigaev et al., 2023). Importantly, using SAE features they are able to manipulate the image generation process in interpretable ways. In contrast, in our work we train SAEs on intermediate representations of the forward pass of SDXL Turbo. Consequently, we can interpret and manipulate SDXL Turbo's forward pass on a finer granularity, e.g., by intervening on specific transformer blocks and spatial positions.

# 7    CONCLUSION

We trained SAEs on SDXL Turbo's (by default) opaque intermediate representations. We highlight that this study is one of the first in the academic literature to mechanistically interpret the intermediate representations of the modern text-to-image model. Our findings demonstrate that SAEs are capable of extracting interpretable features and that they have a significant causal effect on the generated images. Importantly, the learned features shed light on the forward pass of SDXL Turbo. In particular, they enabled us to observe that transformer blocks play a specific and varying role in the generation process. Our findings show a clear picture for the functions of `down.2.1`, `up.0.0`, and, `up.0.1`. For `mid.0`, the picture is less clear; as we observed from its learned features, it seems to encode more abstract information, and interventions are less effective. These observations lead us to the rough hypothesis of how SDXL Turbo generates images: `down.2.1` decides top-level composition, `mid.0` assigns low-level semantics, `up.0.0` adds details based on the two above, and `up.0.1` fills in color, texture, and style.

Although our work provides important insights into the mechanisms of SDXL Turbo, we studied its transformer blocks in isolation. More research is required to understand how the features of SDXL Turbo interact between layers and how this affects the overall functionality of the model. A promising direction would be the application of advanced interpretability techniques such as (Marks et al., 2024). Marks et al. (2024) computes circuits that show how different layers and attention heads wire together and, therefore, would provide insight into our hypothesis stated above.

In addition, the complex nature of some of the learned visual features deserves special attention. Although some features (for example, learning on `down.2.1` and `up.0.1`) exhibit their effect even when turning them on during empty-prompt generations, other features (typically, the ones learned on `mid.0` and `up.0.0`) require an appropriate context to show their effect. This complexity poses additional challenges for the automatic annotation of features. Our preliminary results in this direction suggest that current visual language models do not seem to pick up the features, often subtle roles. Furthermore, we question whether captioning the visual features with few short sentences can adequately capture most features' roles.

We believe that our work highlights the potential of SAEs in revealing the internal structure of diffusion models like SDXL Turbo, and it could help future researchers answer more sophisticated questions about image generation. For example, how does SDXL Turbo add illumination effects, render wool, hair, or reflections of objects in the water?

ETHICS STATEMENT

Our research focuses on enhancing the interpretability of SDXL Turbo using sparse autoencoders with the goal of making complex generative AI systems more transparent and understandable, especially as these models become more advanced and are increasingly used in image generation, content creation, and other influential domains. We acknowledge the ethical responsibilities that come with AI advancements, including the potential misuse of generative models to create harmful content, embedded biases that may exacerbate risks to people—particularly those historically underrepresented or misrepresented in these models. By working on interpretability methods with the goal of increasing our understanding of text-to-image models, we hope to facilitate the identification and mitigation of unintended behaviors or biases within the models. We foresee no particular ethical concerns with the methods developed in this work and hope this paper contributes to developing tools that can identify and mitigate ethical issues in the future.

REPRODUCIBILITY STATEMENT

Upon acceptance we will provide code to reproduce all datasets, experiments, and analyses. In the meantime, we refer to Appendix A for SAE training details and to Appendix F and our supplementary material for further feature visualizations and results.

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

# A  SUPERPOSITION AND SPARSE AUTOENCODERS

Let $h(x) \in \mathbb{R}^d$ be some intermediate result of a forward pass of a neural network on the input $x$. In a fully connected neural network, the components $h(x)$ could correspond to neurons. In transformers, which are residual neural networks with attention and fully connected layers, $h(x)$ usually either refers to the content of the residual stream after some layer or an update to the residual stream by some layer or the neurons within a fully connected block. In general, $h(x)$ could refer to anything, e.g. keys, queries, and values. Bricken et al. (2023) have shown that in many neural networks (especially large language models), intermediate representations can be well approximated by sparse sums of $n_f \in \mathbb{N}$ learned feature vectors, i.e.,

$$h(x) \approx \sum_{\rho=1}^{n_f} s_\rho(x) \mathbf{f}_\rho, \tag{12}$$

where $s_\rho(x)$ are the input dependent (in the literature, input dependence is usually omitted) coefficients of which most are equal to zero and $\mathbf{f}_1, \ldots, \mathbf{f}_{n_f} \in \mathbb{R}^d$ is a learned dictionary of feature vectors.

Importantly, these learned characteristics are usually highly *interpretable* (specific), *sensitive* (fire on the relevant contexts), *causal* (change the output in expected ways in intervention) and usually do not correspond directly to individual neurons. There are also some preliminary results on the universality of these learned features, i.e., that different training runs on similar data result in the corresponding models picking up largely the same features (Bricken et al., 2023).

**Superposition.** By associating task-relevant features with directions in $\mathbb{R}^d$ instead of individual components of $h(x) \in \mathbb{R}^d$, it is possible to represent many more features than there are components, i.e., $n_f >> d$. As a result, in this case, the learned dictionary vectors $\mathbf{f}_1, \ldots, \mathbf{f}_{n_f}$ cannot be orthogonal to each other, which can lead to interference when too many features are on (thus the sparsity requirement). However, it would be theoretically possible to have exponentially (in $d$) many almost orthogonal directions embedded in $\mathbb{R}^d$.[9]

Using representations like this, the optimization process during training can trade off the benefits of being able to represent more features than there are components in $h$ with the costs of features interfering with each other. Such representations are especially effective if the real features underlying the data do not co-occur with each other too much, that is, they are sparse. In other words, in order to represent a single input ("Michael Jordan") only a small subset of the features ("person", ..., "played basketball") is required (Elhage et al., 2022; Bricken et al., 2023).

The phenomenon of neural networks that exploit representations with more features than there are components (or neurons) is called superposition (Elhage et al., 2022). Superposition can explain the presence of polysemantic neurons. The neurons, in this case, are simply at the wrong level of abstraction. The closest feature vector can change when varying a neuron, resulting in the neuron seemingly reacting to or steering semantically unrelated things.

**Sparse autoencoders.** To implement the sparse decomposition from equation 1, the vector $s$ containing the $n_f$ coefficients of the sparse sum, is parameterized by a single linear layer followed by ReLU activations, called the *encoder*,

$$s = \mathrm{ENC}(h) = \sigma(W^{\mathrm{ENC}}(h - b_{\mathrm{pre}}) + b_{\mathrm{act}}), \tag{13}$$

in which $h \in \mathbb{R}^d$ is the latent that we aim decompose, $\sigma(\cdot) = \max(0, \cdot)$, $W^{\mathrm{ENC}} \in \mathbb{R}^{n_f \times d}$ is a learnable weight matrix and $b_{\mathrm{pre}}$ and $b_{\mathrm{act}}$ are learnable bias terms. We omitted the dependencies $h = h(x)$ and $s = s(h)$ that are clear from context.

Similarly, the learnable features are parametrized by a single linear layer, called *decoder*,

$$h' = \mathrm{DEC}(s) = W^{\mathrm{DEC}}s + b_{\mathrm{pre}}, \tag{14}$$

in which $W^{\mathrm{DEC}} = (\mathbf{f}_1 | \cdots | \mathbf{f}_{n_f}) \in \mathbb{R}^{d \times n_f}$ is a learnable matrix of which the columns take the role of learnable features and $b_{\mathrm{pre}}$ is a learnable bias term.

---

[9]It follows from the Johnson-Lindenstrauss Lemma (Johnson et al., 1986) that one can find at least $\exp(d\epsilon^2/8)$ unit vectors in $\mathbb{R}^d$ with the dot product between any two not larger than $\epsilon$.

**Training.** The pair ENC and DEC are trained in a way that ensures that $h'$ is a sparse sum of feature vectors (as in equation 1). Given a dataset of latents $h_1, \ldots, h_n$, both encoder and decoder are trained jointly to minimize a proxy to the loss

$$\min_{\substack{W^{\text{ENC}}, W^{\text{DEC}} \\ b_{\text{pre}}, b_{\text{act}}}} \sum_{i=1}^n \|h_i' - h_i\|_2^2 + \lambda \|s_i\|_0 = \sum_{i=1}^n \|\text{DEC}(\text{ENC}(h_i)) - h_i\|_2^2 + \lambda \|\text{ENC}(h_i)\|_0, \quad (15)$$

where $h_i = h(x_i)$, $s_i = \text{ENC}(h(x_i))$ (when we refer to components of $s$ we use $s_\rho$ instead), the $\|h_i' - h_i\|_2^2$ is a reconstruction loss, $\|s_i\|_0$ a regularization term ensuring the sparsity of the activations and $\lambda$ the corresponding trade-off term.

**Technical details.** In practice, $\|s_i\|_0$ cannot be efficiently optimized directly, which is why it is usually replaced with $\|s_i\|_1$ or other proxy objectives.

In our work, we make use of the top-$k$ formulation from Gao et al. (2024), in which $\|s_i\|_0 \leq k$ is ensured by introducing the a top-$k$ function TopK into the encoder:

$$s = \text{ENC}(h) = \sigma(\text{TopK}(W^{\text{ENC}}(h - b_{\text{pre}}) + b_{\text{act}})). \quad (16)$$

As the name suggests, TopK returns a vector that sets all components except the top $k$ ones to zero.

In addition (Gao et al., 2024) use an auxiliary loss to handle dead features. During training, a sparse feature $\rho$ is considered *dead* if $s_\rho$ remains zero over the last 10M training examples.

The resulting training loss is composed of two terms: the $L_2$-reconstruction loss and the top-auxiliary $L_2$-reconstruction loss for dead feature reconstruction. For a single latent $h$, the loss is defined

$$L(h, h') = \|h - h'\|_2^2 + \alpha \|h - h'_{\text{aux}}\|_2^2 \quad (17)$$

In this equation, the $h'_{\text{aux}}$ is the reconstruction based on the top $k_{\text{aux}}$ dead features. This auxiliary loss is introduced to mitigate the issue of dead features. After the end of the training process, we observed none of them. Following (Gao et al., 2024), we set $\alpha = \frac{1}{32}$ and $k_{\text{aux}} = 256$, performed tied initialization of encoder and decoder, normalized decoder rows after each training step. The number of learned features $n_f$ is set to 5120, which is four times the length of the input vector. The value of $k$ is set to 10 as a good trade-off between sparsity and reconstruction quality. Other training hyperparameters are batch size: 4096, optimizer: Adam with learning rate: $10^{-4}$ and betas: $(0.9, 0.999)$.

# B FEW STEP DIFFUSION MODELS: SDXL TURBO

**Diffusion models.** Diffusion models are a class of generative models that were introduced by Sohl-Dickstein et al. (2015) and are a core component of many of the recent large-scale text-to-image generative models (Ramesh et al., 2022; Rombach et al., 2022; Saharia et al., 2022a). Notably, Ho et al. (2020); Song & Ermon (2020) demonstrates that diffusion model are a viable alternative to GANs (Goodfellow et al., 2014) for image generation. Additionally, diffusion models enjoy stable training dynamics, are easier to scale than GANs (Dhariwal & Nichol, 2021), and offer likelihood estimates of samples (Song et al., 2021).

Diffusion models sample from an unknown distribution $p$ by learning to iteratively denoise corrupted samples, starting from pure noise. The corruption process is defined on training samples from $p$. Mathematically, the images are corrupted with Gaussian noise and are distributed according to

$$q_t(x_t|x_0) := \mathcal{N}(\alpha_t x_0, \sigma_t^2 \mathbf{I}), \quad (18)$$

where $x_0$ corresponds to a real image from $p$, $0 \leq t \leq T$, $\alpha_t, \sigma_t^2$ are positive real-valued scalars such that the signal-to-noise ratio $SNR := \frac{\alpha_t}{\sigma_t^2}$ is monotonically decreasing. Additionally, the coefficients $\alpha_{T-1}, \sigma_{T-1}^2$ are typically chosen such that $x_T \sim \mathcal{N}(0, \mathbf{I})$. In this work, the number of corruption steps $T$ is fixed to 1000, as we study the pre-trained models from (Sauer et al., 2023b). Given this predetermined corruption process, the diffusion model learns to reverse it to recover clean data.

The denoising process is implemented via a distribution $p_\theta(x_{t-1}|x_t)$. The simplest way to generate samples using $p_\theta(x_{t-1}|x_t)$ is to first generate a sample of pure noise $x_T \sim \mathcal{N}(0, \mathbf{I})$, followed by $T$

iterative applications of $p_\theta$, yielding a sequence $x_T, x_{T-1}, ..., x_1, x_0$, where $x_0$ approximates samples from $p$. The vector $\theta$ represents the parameters of a neural network that defines $p_\theta(x_{t-1}|x_t)$. There exist many objectives to learn to reverse the corruption process (Ho et al., 2020; Kingma et al., 2023; Song & Ermon, 2020), but $p_\theta$ is generally trained to minimize the Kullback-Leibler divergence between adjacent steps of the corruption process: $D_{KL}[q_t(x_{t-1}|x_t, x_0)||p_\theta(x_{t-1}|x_t)]$ for $t \in \{1, ..., T\}$, where $q_t(x_{t-1}|x_t, x_0)$ is a Gaussian distribution whose mean and variance can be computed in closed-form using Bayes rule and the definition in eq. (18) (Ho et al., 2020). The denoising distribution $p_\theta(x_{t-1}|x_t)$ is parameterized to be Gaussian. The Kullback-Leibler divergence between two Gaussians admits a simple closed-form solution (Duchi John, 2020), hence, the objective can be efficiently implemented.

The neural network used to parameterize $p_\theta(x_{t-1}|x_t)$ can be trained to learn different quantities (Luo, 2022; Salimans & Ho, 2022; Karras et al., 2022). A possible approach is to directly output the mean $\mu_t$ of $p_\theta(x_{t-1}|x_t)$, while the variance is either fixed or learned as well. In this work, the neural network is parameterized to predict the noise added to the original sample during the forward process (eq. (18)). This is achieved by minimizing the objective $w(t)\|\epsilon - \epsilon_\theta(\alpha_t x_0 + \sigma_t \epsilon, t)\|^2$, where $w$ is a weighting function (Ho et al., 2020). Once $\epsilon_\theta$ is trained, the mean of $p_\theta(x_{t-1}|x_t)$ is computed as $\frac{1}{\alpha_t}(x_t - \sigma_t \epsilon_\theta)$ (Rombach et al., 2022). Since our primary goal is to analyze a pre-trained diffusion model, we refer the interested reader to Rombach et al. (2022); Luo (2022); Salimans & Ho (2022) for more details.

**Latent diffusion.** Originally, diffusion models operated directly on pixels (Ho et al., 2020; Song & Ermon, 2020). However, training a denoising network in pixel space is difficult and expensive (Hoogeboom et al., 2023). As such, Rombach et al. (2022) uses a pre-trained auto-encoder to first compress images, similar to VQGAN (Esser et al., 2021), and define a diffusion process in the latent space instead of the pixel space. To make this difference clear they write $p_\theta(z_{t-1}|z_t)$, in which now $z_t$ refers to a noisy latent instead of a noisy image.

**Distilled diffusion for fast inference.** To speed-up inference of latent diffusion models, Sauer et al. (2023b) distills a pre-trained Stable Diffusion XL (SDXL) model (Podell et al., 2023). The distilled model is referred to as *SDXL Turbo* as it allows high-quality sampling in as little as 1-4 steps. The original SDXL model is trained with a noise schedule of 1000 steps, but in practice, sampling with 20 to 50 steps still generates high-quality images. The speed-up in SDXL Turbo is achieved through a combination of two objectives. First, Sauer et al. (2023b) defines an adversarial game, similar to GANs (Goodfellow et al., 2014). The discriminator is implemented using lightweight classification heads on top of frozen features extracted at $K$ different layers of a DINOv2 backbone (Oquab et al., 2024). Concretely, the objective of the discriminator is given by

$$\mathcal{L}_{\text{adv}}^D = \mathbb{E}_{x_0}\left[\sum_{k=1}^{K}(1 - \mathcal{D}_k(F_k(x_0)))_+ + \gamma R1(\phi)\right] + \mathbb{E}_{\hat{x}_\theta}\left[\sum_{k=1}^{K}(1 + \mathcal{D}_k(F_k(\hat{x}_\theta)))_+\right], \quad (19)$$

where $(x)_+ = \max(0, x)$ is the positive part, $F_k$ denotes the $k$-th features tensor from the DINOv2 backbone, $\mathcal{D}_k$ the $k$-th classification head, $\phi$ is the discriminator parameters, $R1$ is an $L^2$ penalty term on the norm of the gradients, introduced by Mescheder et al. (2018) and $\gamma$ is a scalar hyperparameter. Instead of a traditional classification loss, Sauer et al. (2023b) use the hinge loss (Lim & Ye, 2017), following Sauer et al. (2021; 2023a).

Finally, $\hat{x}_\theta$ represents the prediction of the diffusion model being distilled for the ground-truth image $x_0$ given a noisy sample $x_t$ obtained through the forward diffusion process defined in eq. (18). Sauer et al. (2023b) found that distilling a diffusion model using the adversarial objective only resulted in a model with a FID of 20.8. To further improve performance, they also distilled the noise predictions of the teacher model. Importantly, both the teacher and student models were initialized with the same pre-trained weights. After adversarial distillation (Sauer et al., 2023b), the model learns to map noise to samples in one step

**Neural network architecture.** The denoising network of *SDXL Turbo* estimating $p_\theta(z_{t-1}|z_t)$ is implemented using a U-net similar to Rombach et al. (2022). The U-net is composed of a down-sampling path, a bottleneck, and an up-sampling path. Both the down-sampling and up-sampling paths are composed of 3 individual blocks. The individual block structure differs slightly but both down- and up-sampling blocks consist of residual layers as well as cross-attention transformer blocks. Finally, the bottleneck layer is also composed of attention and residual layers. As for the

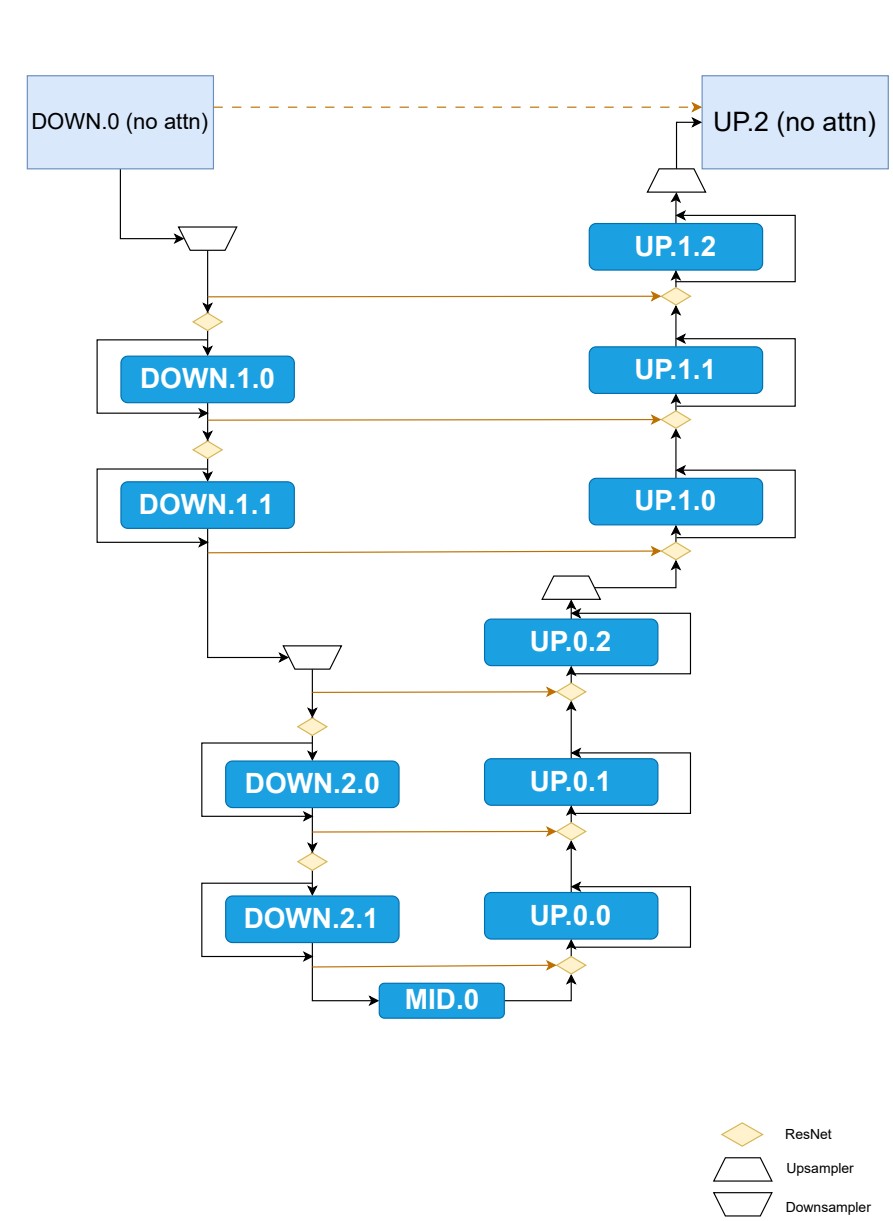

Figure 2: Cross-attention transformer blocks in SDLX's U-net.

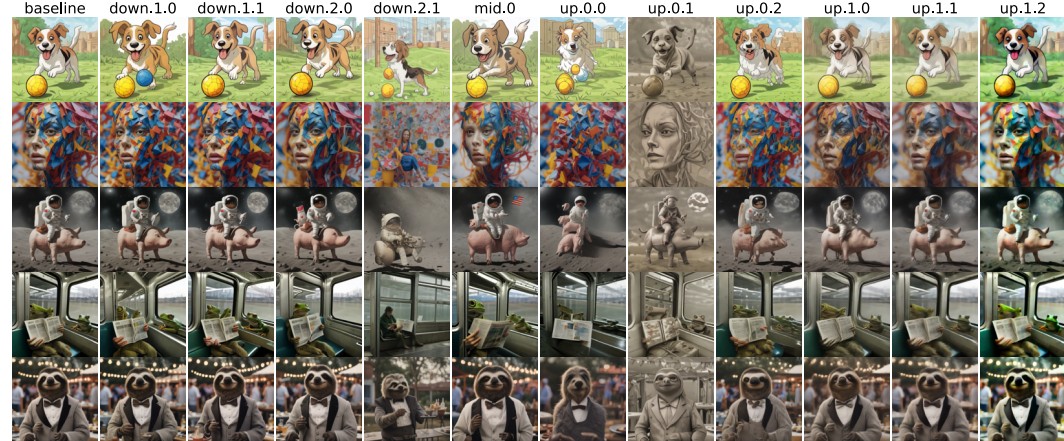

Figure 3: We generate images for the prompts "A dog playing with a ball cartoon.", "A photo of a colorful model.", "An astronaut riding on a pig on the moon.", "A photograph of the inside of a subway train. There are frogs sitting on the seats. One of them is reading a newspaper. The window shows the river in the background." and "A cinematic shot of a professor sloth wearing a tuxedo at a BBQ party." while ablating the updates performed by different cross-attention layers (indicated by the titles). The title "baseline" corresponds to the generation without interventions.

original U-net architecture (Ronneberger et al., 2015), the corresponding blocks in the up-sampling and and down-sampling path are connected via a skip connection. Importantly, the text conditioning is achieved via cross-attention to text embeddings performed by in total 11 transformer blocks embedded in the down-, up-sampling paths and bottleneck. An architecture diagram displaying the relevant blocks can be found in App. B Fig. 2.

## C  FINDING CAUSALLY IMPACTFUL TRANSFORMER BLOCKS

As a first step, we narrow down design space of the 11 cross-attention transformer blocks (see Fig. 2) to those with the highest causal impact on the output. In order to assess their causal impact on the output we qualitatively study the effect of individually ablating each of them (see Fig. 3). As can be seen in Fig. 3 each of the middle blocks down.2.1, mid.0, up.0.0, up.0.1 have a relatively high impact on the output respectively. In particular, the blocks down.2.1 and up.0.1 stand out. It seems like most colors and textures are added in up.0.1, which in the community is already known as "style" block Spinelli (2024). Ablating down.2.1, which is also already known in the community as "composition" block, impacts the entire image composition, including object sizes, orientations and framing. The effects of ablating other blocks such as mid.0 and up.0.0 are more subtle. For mid.0 it is difficult to describe in words and up.0.0 seems to add local details to the image while leaving the overall composition mostly intact.

## D  SAE TRAINING RESULTS

We trained several SAEs with different sparsity levels and sparse layer sizes and observed no dead features. To assess reconstruction quality, we processed 100 random LAION-COCO prompts through a one-step SDXL Turbo process, replacing the additive component of the corresponding transformer block with its SAE reconstruction.

The explained variance ratio and the output effects caused by reconstruction are shown in Table 2. Fig. 4 presents random examples of reconstructions from an SAE with the following hyperparameters: $k = 10, n_f = 5120$, trained on down.2.1. The reconstruction causes minor deviations in the images, and the fairly low LPIPS (Zhang et al., 2018) and pixel distance scores also support these findings. However, to prevent these minor reconstruction errors from affecting our analysis of interventions, we decided to directly add or subtract learned directions from dense feature maps.

Table 2: Distances and explained variance ratio in generated images. "Mean" represents the average pixel Manhattan distance between original and reconstruction-intervened images, with a maximum possible value of 765. "Median" represents the median Manhattan distance per pixel, averaged over all images. 'LPIPS' refers to the average LPIPS score, measuring perceptual similarity. "Explained variance ratio" denotes the ratio of variance explained by the trained SAEs to the total variance.

| $k$ | $n_f$ | Configuration | Mean \| Median | LPIPS | Explained Variance Ratio (%) |
|---|---|---|---|---|---|
| 5 | 640 | down | 83.29 \| 50.04 | 0.3383 | 56.0 |
| | | mid | 52.64 \| 26.82 | 0.2032 | 43.4 |
| | | up0 | 55.89 \| 30.69 | 0.2276 | 44.8 |
| | | up | 52.67 \| 34.53 | 0.2073 | 50.3 |
| | 5120 | down | 74.68 \| 41.49 | 0.3036 | 67.8 |
| | | mid | 48.82 \| 24.60 | 0.1845 | 50.8 |
| | | up0 | 49.19 \| 25.86 | 0.1969 | 57.2 |
| | | up | 47.50 \| 31.11 | 0.1775 | 59.5 |
| 10 | 640 | down | 73.65 \| 41.79 | 0.2893 | 62.8 |
| | | mid | 46.80 \| 23.10 | 0.1772 | 51.5 |
| | | up0 | 48.43 \| 25.80 | 0.1908 | 52.5 |
| | | up | 43.06 \| 26.85 | 0.1638 | 58.7 |
| | 5120 | down | 64.97 \| 34.77 | 0.2582 | 73.7 |
| | | mid | 44.02 \| 21.72 | 0.1627 | 58.8 |
| | | up0 | 42.08 \| 21.54 | 0.1624 | 64.2 |
| | | up | 39.77 \| 24.84 | 0.1453 | 67.1 |
| 20 | 640 | down | 59.29 \| 31.47 | 0.2291 | 69.9 |
| | | mid | 39.95 \| 19.44 | 0.1459 | 60.0 |
| | | up0 | 40.15 \| 21.06 | 0.1499 | 60.9 |
| | | up | 31.97 \| 18.15 | 0.1196 | 66.7 |
| | 5120 | down | 56.37 \| 29.04 | 0.2190 | 78.8 |
| | | mid | 37.28 \| 17.82 | 0.1328 | 66.5 |
| | | up0 | 35.73 \| 18.03 | 0.1302 | 70.6 |
| | | up | 30.31 \| 17.22 | 0.1104 | 74.2 |

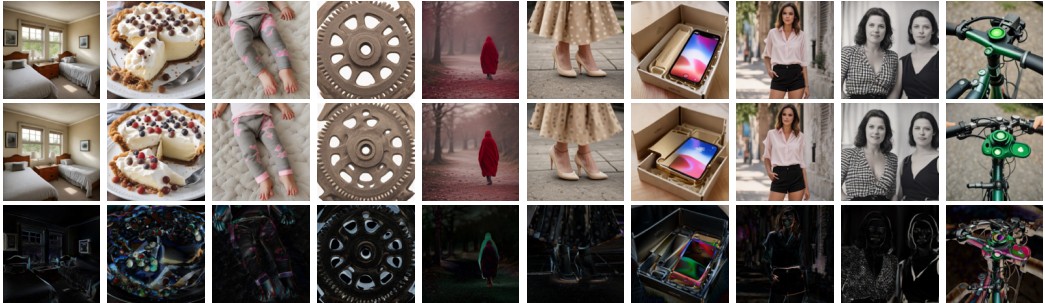

Figure 4: Images generated from 10 random prompts taken from the LAION-COCO dataset are shown in the first row. In the second row, down.2.1 updates are replaced by their SAE reconstructions ($k = 10, n_f = 5120$). The third row visualizes the differences between the original and reconstructed images.

# E  CASE STUDY I: MOST ACTIVE FEATURES ON A PROMPT

Here we provide a version of the case study in Sec. 4.2 but with the top-9 features.

Combining all our feature visualization techniques, in Fig. 5, we depict the features with the highest average activation when processing the prompt: "A cinematic shot of a professor sloth wearing a tuxedo at a BBQ party". We discuss each transformer block, sorted from easiest to hardest to interpret.

**Down.2.1.** We start with down.2.1, which indeed seems to contribute towards the image composition. Several features seem to relate directly to phrases of the prompt: 4539 "professor sloth", 4751, 1226, "wearing a tuxedo", 2881, 567, 3119, 2345 "party".

Turning off features (A. -6.0 column) removes elements from and changes elements in the scene in ways that mostly make sense when comparing with the heatmap (hmap column) and the top examples (C columns): 1674 *removes* the light chains in the back, 4608 the umbrellas/tents, 4539 the 3D animation-like sloth face, 567 people in the background, 3119, 2345 some of the light chains, and, 4751 *changes* the type of suit, 1226 the shirt. Similarly, enhancing the same features (A. 6.0 column) enhances the corresponding elements and sometimes changes them.

Activating the features on the empty prompt often creates related elements. Note that, for the fixed random seed we use, the empty prompt itself looks like a painting of a piece of nature with a lot of green and brown. Therefore, while the prompt is empty the features active during the forward pass are not and due to the layers that we don't intervene on still contribute to the images.

While top dataset examples (C.0, C.1 columns) and also empty prompt intervention (B. column) mostly agree with the feature activation heatmaps (hmap column), some of them add additional insight, e.g., 2881, which activates on the suit, seems to correspond to (masqueraded) characters in a (festive) scene, 3119 seems to be about party decorations in general and not just light chains, 2345 seems to react to other celebration backgrounds as well.

**Up.0.1.** The up.0.1 transformer block indeed seems to contribute substantially to the style of the image. They are hard to relate directly to phrases in the prompt, yet indirectly they do relate. E.g., the illumination (2727) and shadow (500, 1700) effects probably have something to do with "a cinematic shot" and the animal hair texture (2314) with "sloth". Beyond that several features seem to mainly contribute to the glowing lights in the background (1295, 4238, 2341).

Interestingly, turning on the up.0.1 features on the entire empty prompt (B. column) results in texture-like images. In contrast, when activating them locally (A. columns) their contribution to the output is highly localized and keeps most of the remaining image largely unchanged. For the up.0.1 we find it remarkable that often the ablation and amplification are counterparts: 500 (light, shadow), 2727 (shadow, light), 3936 (blue, orange), 2314 (less grey hair, more brown hair).

**Up.0.0.** To the best of our knowledge, up.0.0's responsibility is less understood. First, we observe that it acts very locally and we think that it often requires relevant other features from the previous and subsequent transfomrer blocks effectively influence the image. Turning it on the empty prompt results in abstract looking images, which are hard to relate to the other columns, which is why we excluded this visualization technique for this transformer block and instead included an additional example.

Most top dataset examples and their activations (C columns) are highly interpretable: 3603 party decoration, 5005 upper part of tent, 775 buttons on suit, 153 lower animal jaw, 1550 collars, 2648 pavilions, 1604 right part of the image, 564 bootie. Many of the features have a expected causal effect on the generation when ablating/enhancing (B. columns): 3603, 5005, 775, 153, 1550, 564, but not all: 2221, 2648, 1604. To sum up, this transformer block seems to mostly add local details to the generation and when interventions are performed locally they are effective.

**Mid.0.** Again to the best of our knowledge, mid.0's role is also not well understood. We find it harder to interpret because most interventions on the mid.0 have very subtle effects. Again, we did not include the intervention in which we turn on the features on the empty prompt because interventions of this kind in mid.0 barely affect the generation.

While effects of interventions are subtle, dataset examples (C. columns) and heatmap (hmap column) all mostly agree with each other and are specific enough to be interpretable: 4755 bottom right part

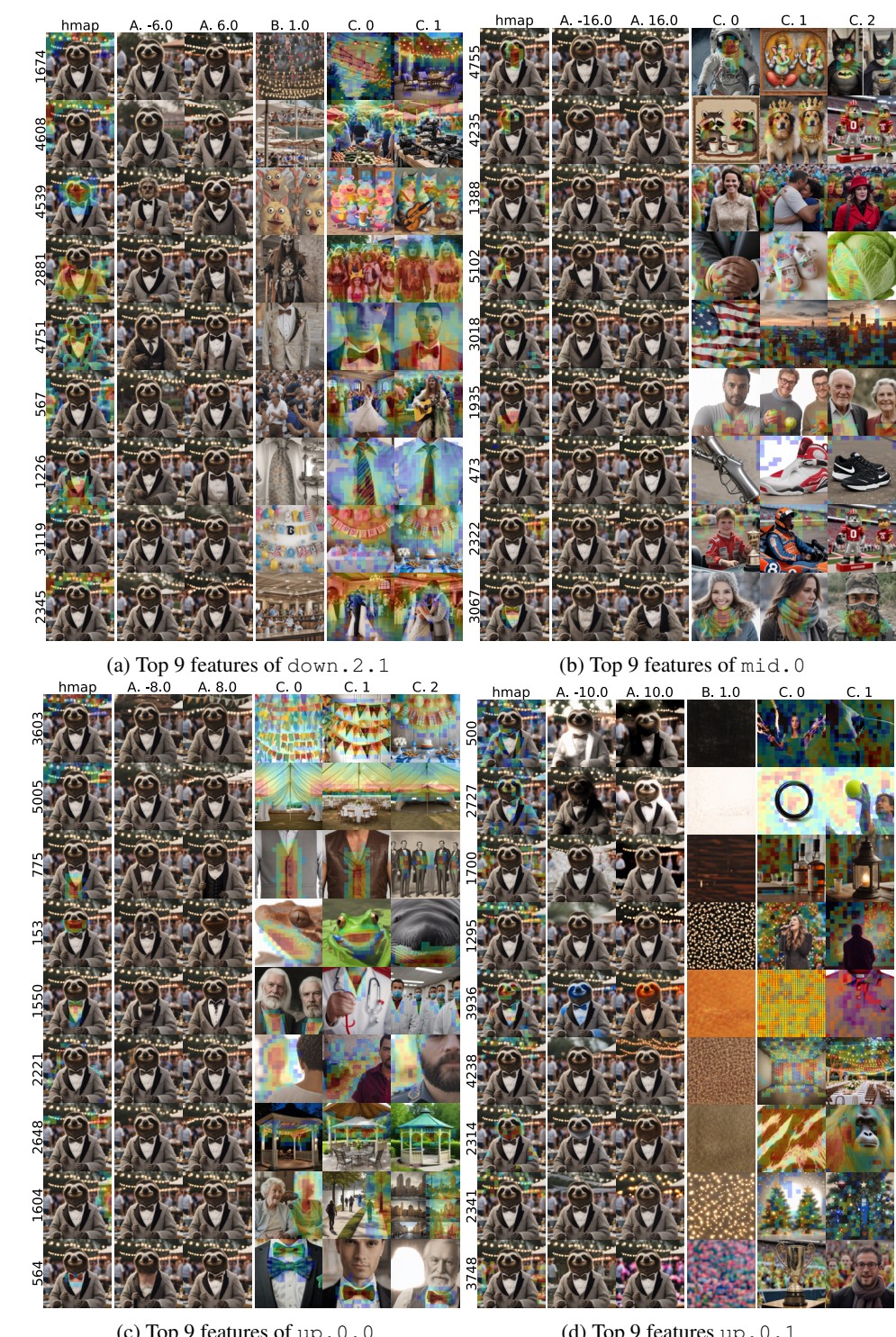

(a) Top 9 features of down.2.1

(b) Top 9 features of mid.0

(c) Top 9 features of up.0.0

(d) Top 9 features up.0.1

Figure 5: The top 9 features of down.2.1 (a), mid.0 (b), up.0.0 (c) and up.0.1 (d) for the prompt: "A cinematic shot of a professor sloth wearing a tuxedo at a BBQ party." Each row represents a feature. The first column depicts a feature heatmap (highest activation red and lowest nonzero one blue). The column titles containing "A" show feature modulation interventions, the ones containing "B" the intervention of turning on the feature on the empty prompt, and the ones containing "C" depict top dataset examples. Floating point values in the title denote $\beta$ and $\gamma$ values.

of faces, 4235 left part of (animal) faces, 1388 people in the background, 1935 is active on chests, 473 mostly active on the image border, 2322 again seems to have to do with backgrounds that also contain people, 3067 active on the neck or neck accessories, and, 5102 outlines the left border of the main object in the scene. The feature 3018 is difficult to interpret.

Our observations indicate that `mid.0`'s features are more abstract, indicate where things are[10] and potentially how they relate to each other.

## F  FEATURES INCLUDING PROMPTS

**Feature plots.** We provide the prompts for Fig. 6 in Table 3. Further we provide the same plots for the last six feature indices of each transformer block in Fig. 7 and the corresponding prompts in Table 4. Additionally, provide some cherry picked features for `down.2.1` and `up.0.1` in Fig. 8 and the corresponding prompts in Table 5.

**Intervention plots.** Additionally, we provide plots in which we turn on features from Fig. 8 but in unrelated prompts (as opposed to top dataset example prompts that already activate the features by themselves). For simplicity here we simply turn on the features across all spatial locations, which does not seem to be a well suitable strategy for `up.0.1`, which usually acts locally. To showcase, the difference we created one example image in Fig. 10, in which we manually draw localized masks to turn on the corresponding features.

## G  ANNOTATION PIPELINE DETAILS

We used GPT-4o to caption learned features on `down.2.1`. For each feature, the model was shown a series of 5 unrelated images, a progression of 9 images, the $i$-th of those corresponds to $\sim i \cdot 10\%$ average activation value of the maximum. Finally, we show 5 images corresponding to the highest average activations. Since some features are active on particular parts of images, the last 9 images are provided alongside their so-called "coldmaps": a version of an image with weakly active and inactive regions being faded and concealed.

The images were generated by 1-step SDXL Turbo diffusion process on $50'000$ random prompts of LAION-COCO dataset.

### G.1  TEXTUAL PROMPT TEMPLATE

Here is the prompt template for the VLM.

> **System.** You are an experienced mechanistic interpretability researcher that is labeling features from the hidden representations of an image generation model.
>
> **User.** You will be shown a series of images generated by a machine learning model. These images were selected because they trigger a specific feature of a sparse auto-encoder, trained to detect hidden activations within the model. This feature can be associated with a particular object, pattern, concept, or a place on an image. The process will unfold in three stages:
>
> 1. **Reference Images:** First, you'll see several images *unrelated* to the feature. These will serve as a reference for comparison.
>
> 2. **Feature-Activating Images:** Next, you'll view images that activate the feature with varying strengths. Each of these images will be shown alongside a version where non-activated regions are masked out, highlighting the areas linked to the feature.
>
> 3. **Strongest Activators:** Finally, you'll be presented with the images that most strongly activate this feature, again with corresponding masked versions to emphasize the activated regions.

---

[10]SDXL Turbo does not utilize positional encodings for the spatial locations in the feature maps. Therefore, we did a brief sanity check and trained linear probes to detect $i, j$ given $D_{ij}^{in}$. These probes achieved high accuracy on a holdout set: $97.9\%, 98.48\%, 99.44\%, 95.57\%$ for `down.2.1, mid.0, up.0.0, up.0.1`.

Your task is to carefully examine all the images and identify the thing or concept represented by the feature. Here's how to provide your response:

- **Reasoning:** Between '<thinking>' and '</thinking>' tags, write up to 400 words explaining your reasoning. Describe the visual patterns, objects, or concepts that seem to be consistently present in the feature-activating images but not in the reference images.

- **Expression:** Afterward, between '<answer>' and '</answer>' tags, write a concise phrase (no more than 15 words) that best captures the common thing or concept across the majority of feature-activating images.

Note that not all feature-activating images may perfectly align with the concept you're describing, but the images with stronger activations should give you the clearest clues. Also pay attention to the masked versions, as they highlight the regions most relevant to the feature.

**User.** These images are not related to the feature: Reference Images

**User.** This is a row of 9 images, each illustrating increasing levels of feature activation. From left to right, each image shows a progressively higher activation, starting with the image on the far left where the feature is activated at 10% relative to the image that activates it the most, all the way to the far right, where the feature activates at 90% relative to the image that activates it the most. This gradual transition highlights the feature's growing importance across the series. {Feature-Activating Images}

**User.** This row consists of 9 masked versions of the original images. Each masked image corresponds to the respective image in the activation row. Areas where the feature is not activated are completely concealed by a white mask, while regions with activation remain visible.) {Feature-Activating Images Coldmaps}

**User.** These images activate the feature most strongly. {Strongest Activators}

**User.** These masked images highlight the activated regions of the images that activate the feature most strongly. The masked images correspond to the images above. The unmasked regions are the ones that activate the feature. {Strongest Activators Coldmaps}

## G.2  EXAMPLE OF PROMPT IMAGES

The images used to annotate feature 0 are shown in Fig. 11.

## G.3  EXAMPLES OF GENERATED CAPTIONS

We present the captions generated by GPT-4o for the first and last 10 features in Table 6.

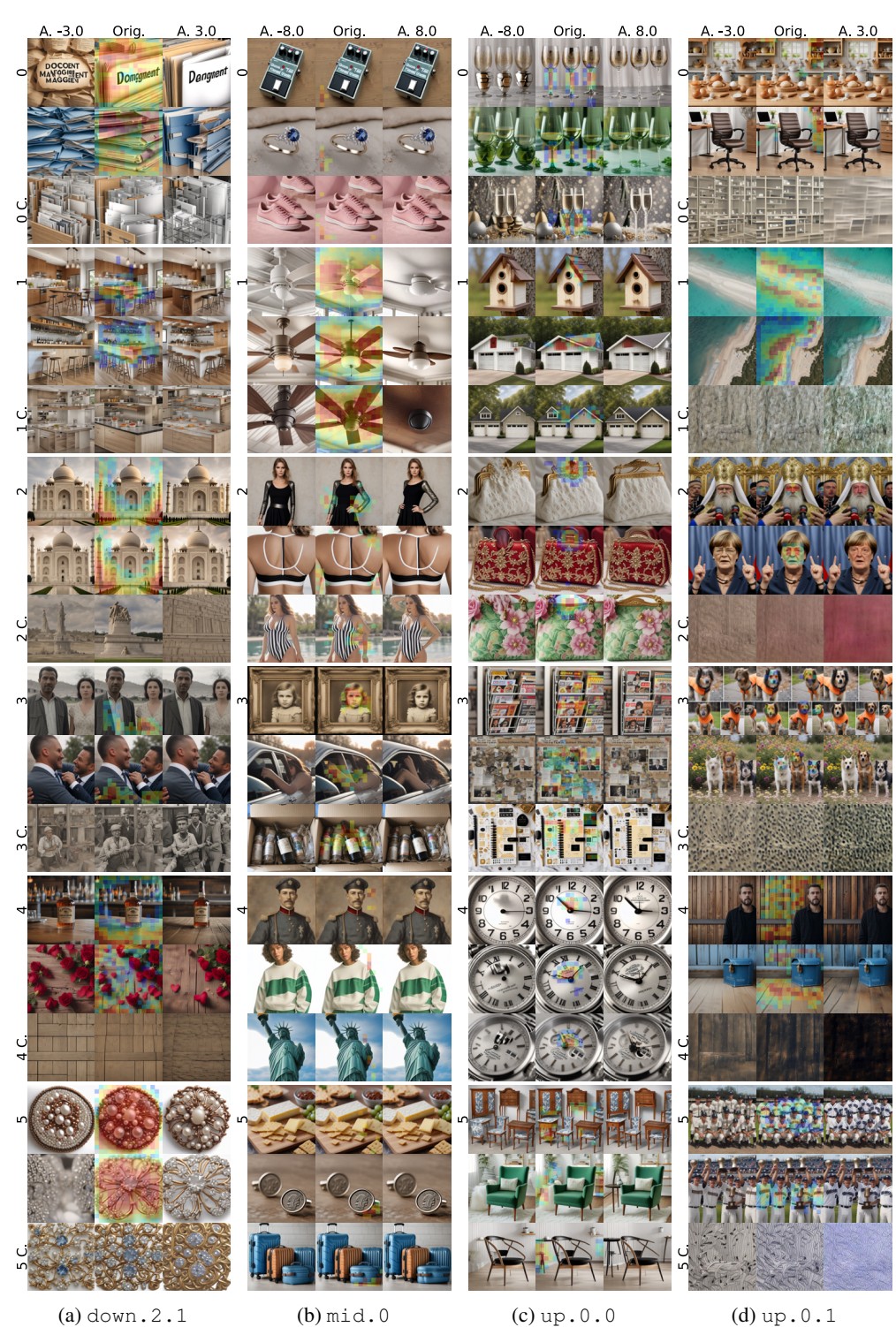

Figure 6: We visualize 6 features for `down.2.1` (a), `mid.0` (b), `up.0.0`, and `up.0.1`. We use three columns for each transformer block and three rows for each feature. For `down.2.1` and `up.0.1` we visualize the two samples from the top 5% quantile of activating dataset examples (middle) together a feature ablation (left) and a feature enhancement (right), and, activate the feature on the empty prompt with $\gamma = 0.5, 1, 2$ from left to right. For `mid.0` and `up.0.0` we display three samples with ablation and enhancement. Captions are in Table 3.

Table 3: Prompts for the top 5% quantile examples in Fig. 6

| Block | Feature | Prompt |
|---|---|---|
| down.2.1 | 0 | A file folder with the word document management on it. |
| | 0 | Two blue folders filled with dividers. |
| | 1 | A kitchen with an island and bar stools. |
| | 1 | An unfinished bar with stools and a wood counter. |
| | 2 | The Taj Mahal, or a white marble building in India. |
| | 2 | The Taj Mahal, or a white marble building in India. |
| | 3 | A man and woman standing next to each other. |
| | 3 | Two men in suits hugging each other outside. |
| | 4 | An old Forester whiskey bottle sitting on top of a wooden table. |
| | 4 | Red roses and hearts on a wooden table. |
| | 5 | A beaded brooch with pearls and copper. |
| | 5 | An image of a brooch with diamonds. |
| mid.0 | 0 | The Boss TS-3W pedal has an electronic tuner. |
| | 0 | An engagement ring with blue sapphire and diamonds. |
| | 0 | The women's pink sneaker is shown. |
| | 1 | A white ceiling fan with three blades. |
| | 1 | A ceiling fan with three blades and a light. |
| | 1 | The ceiling fan is dark brown and has two wooden blades. |
| | 2 | The black dress is made from knit and has metallic sleeves. |
| | 2 | The back view of a woman wearing a black and white sports bra. |
| | 2 | The woman is wearing a striped swimsuit. |
| | 3 | An old-fashioned photo frame with a little girl on it. |
| | 3 | The woman is sitting in her car with her head down. |
| | 3 | The contents of an empty bottle in a box. |
| | 4 | An old painting of a man in uniform. |
| | 4 | The model wears an off-white sweatshirt with green panel. |
| | 4 | The Statue of Liberty stands tall in front of a blue sky. |
| | 5 | Cheese and crackers on a cutting board. |
| | 5 | Two cufflinks with coins on them. |
| | 5 | Three pieces of luggage are shown in blue. |
| up.0.0 | 0 | Three wine glasses with gold and silver designs. |
| | 0 | Three green wine glasses sitting next to each other. |
| | 0 | New Year's Eve with champagne, gold, and silver. |
| | 1 | The birdhouse is made from wood and has a brown roof. |
| | 1 | The garage is white with red shutters. |
| | 1 | Two garages with one attached porch and the other on either side. |
| | 2 | An elegant white lace purse with gold clasp. |
| | 2 | The red handbag has gold and silver designs. |
| | 2 | A pink and green floral-colored purse. |
| | 3 | A magazine rack with magazines on it. |
| | 3 | The year-in-review page for this digital scrap. |
| | 3 | The planner sticker kit is shown with gold and black accessories. |
| | 4 | A clock with numbers on the face. |
| | 4 | A silver watch with roman numerals on the face. |
| | 4 | An automatic watch with a silver dial. |
| | 5 | Four pieces of wooden furniture with blue and white designs. |
| | 5 | The green chair is in front of a white rug. |
| | 5 | The wish chair with a black seat. |
| up.0.1 | 0 | The wooden toy kitchen set includes bread, eggs, and flour. |
| | 0 | The office chair is brown and black. |
| | 1 | An aerial view of the white sand and turquoise water. |
| | 1 | An aerial view of the beach and ocean. |
| | 2 | The patriarch of Ukraine is shown speaking to reporters. |
| | 2 | German Chancellor Merkel gestures as she speaks to the media. |
| | 3 | Four pictures showing dogs wearing orange vests. |
| | 3 | Two dogs are standing on the ground next to flowers. |
| | 4 | A man standing in front of a wooden wall. |
| | 4 | A blue mailbox sitting on top of a wooden floor. |
| | 5 | The baseball players are posing for a team photo. |
| | 5 | The baseball players are holding up their trophies. |

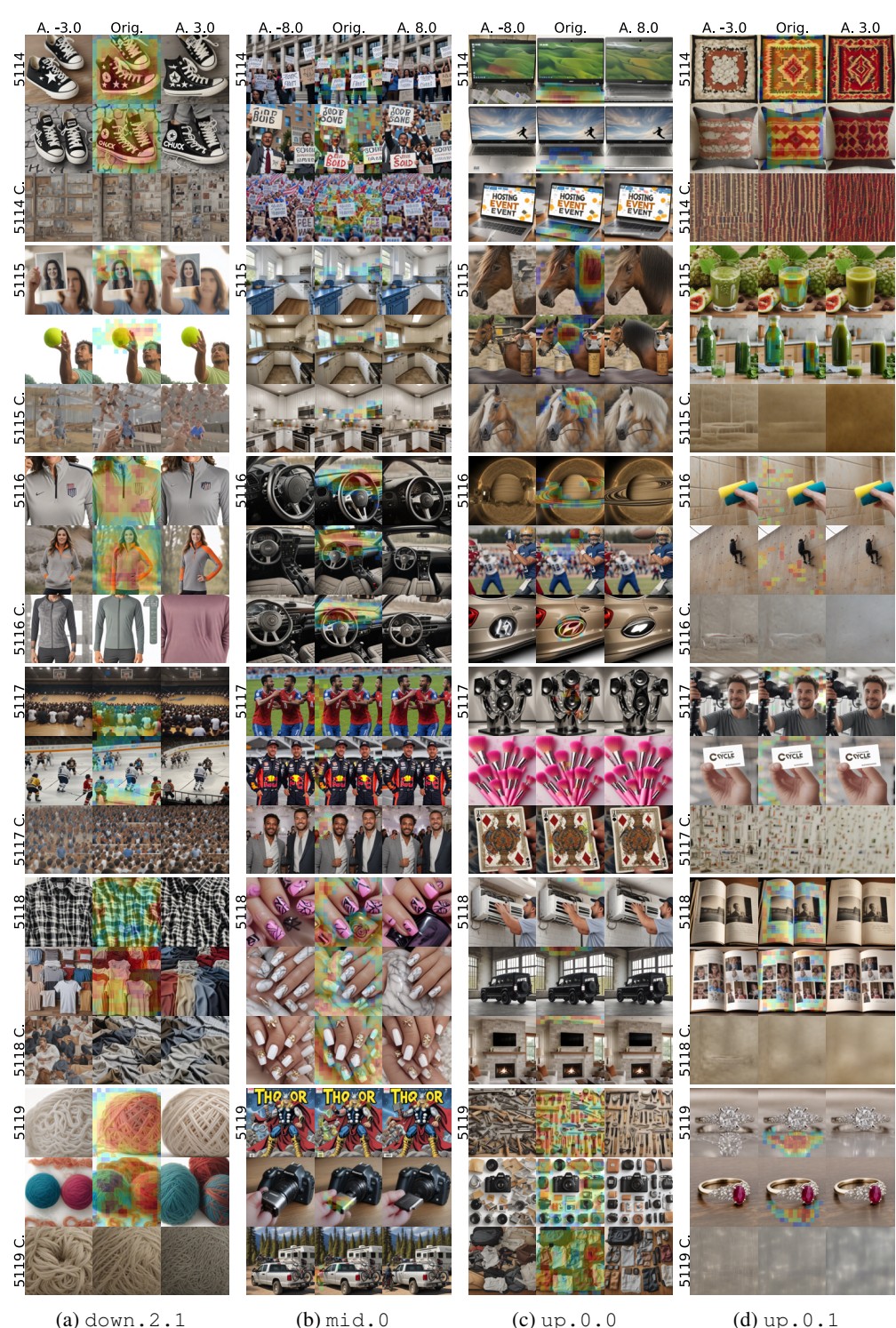

Figure 7: We visualize last 6 features for down.2.1 (a), mid.0 (b), up.0.0, and up.0.1. We use three columns for each transformer block and three rows for each feature. For down.2.1 and up.0.1 we visualize two samples from the top 5% quantile of activating dataset examples (middle) together a feature ablation (left) and a feature enhancement (right), and, activate the feature on the empty prompt with $\gamma = 0.5, 1, 2$ from left to right. For mid.0 and up.0.0 we display three samples with ablation and enhancement. Captions are in Table 4.

Table 4: Prompts for the top 5 % quantile examples in Fig. 7

| Block | Feature | Prompt |
|---|---|---|
| down.2.1 | 5114 | Black and white Converse sneakers with the word black star. |
| | 5114 | Black and white Converse sneakers with the word Chuck. |
| | 5115 | A woman holding up a photo of herself. |
| | 5115 | A man holding up a tennis ball in the air. |
| | 5116 | The Nike Women's U.S. Soccer Team DRI-Fit 1/4 Zip Top. |
| | 5116 | The women's gray and orange half-zip sweatshirt. |
| | 5117 | A large group of people sitting in front of a basketball court. |
| | 5117 | Hockey players are playing in an arena with spectators. |
| | 5118 | The black and white plaid shirt is shown. |
| | 5118 | The different colors and sizes of t-shirts. |
| | 5119 | A ball of yarn on a white background. |
| | 5119 | Two balls of colored wool are on the white surface. |
| mid.0 | 5114 | People holding signs in front of a building. |
| | 5114 | Two men dressed in suits and ties are holding up signs. |
| | 5114 | A large group of people holding flags and signs. |
| | 5115 | A kitchen with white cabinets and a blue stove. |
| | 5115 | The kitchen is clean and ready for us to use. |
| | 5115 | A kitchen with white cabinets and stainless steel appliances. |
| | 5116 | The steering wheel and dashboard in a car. |
| | 5116 | The interior of a car with dashboard controls. |
| | 5116 | The dashboard and steering wheel in a car. |
| | 5117 | Three men are celebrating a goal on the field. |
| | 5117 | Two men in Red Bull racing gear standing next to each other. |
| | 5117 | Two men are posing for the camera at an event. |
| | 5118 | Someone is holding up their nail polish with pink and black designs. |
| | 5118 | The nail is very cute and looks great with marble. |
| | 5118 | White stily nails with gold and diamonds. |
| | 5119 | The Mighty Thor comic book. |
| | 5119 | The camera is showing its flash drive. |
| | 5119 | A truck with bikes on the back parked next to a camper. |
| up.0.0 | 5114 | The Acer laptop is open and ready to use. |
| | 5114 | The Lenovo S13 laptop is open and has an image of a person jumping off the keyboard. |
| | 5114 | A laptop with the words Hosting Event on it. |
| | 5115 | A horse with a black nose and brown mane. |
| | 5115 | The horse leather oil is being used to protect horses. |
| | 5115 | An oil painting on a canvas of a horse. |
| | 5116 | The sun is shining brightly over Saturn. |
| | 5116 | A football player throws the ball to another team. |
| | 5116 | Car door light logo sticker for Hyundai. |
| | 5117 | An artistic black and silver sculpture with speakers. |
| | 5117 | The pink brushes are sitting on top of each other. |
| | 5117 | Four kings playing cards in the hand. |
| | 5118 | A man is fixing an air conditioner. |
| | 5118 | The black Land Rover is parked in front of a large window. |
| | 5118 | A flat screen TV mounted on the wall above a fireplace. |
| | 5119 | A table with many different tools on it. |
| | 5119 | A camera with many different items including flash cards, lenses, and other accessories. |
| | 5119 | The contents of an open suitcase and some clothes. |
| up.0.1 | 5114 | An old Navajo rug with multicolored designs. |
| | 5114 | The pillow is made from an old kilim. |
| | 5115 | An image of noni juice with some fruits. |
| | 5115 | A bottle and glass on the counter with green juice. |
| | 5116 | Someone cleaning the shower with a sponge. |
| | 5116 | A man on a skateboard climbing a wall with ropes. |
| | 5117 | A man taking a selfie in front of some camera equipment. |
| | 5117 | A person holding up a business card with the words cycle transportation. |
| | 5118 | Two photos are placed on top of an open book. |
| | 5118 | An open book with pictures of children and their parents. |
| | 5119 | An engagement ring with diamonds on top. |
| | 5119 | An oval ruby and diamond ring. |

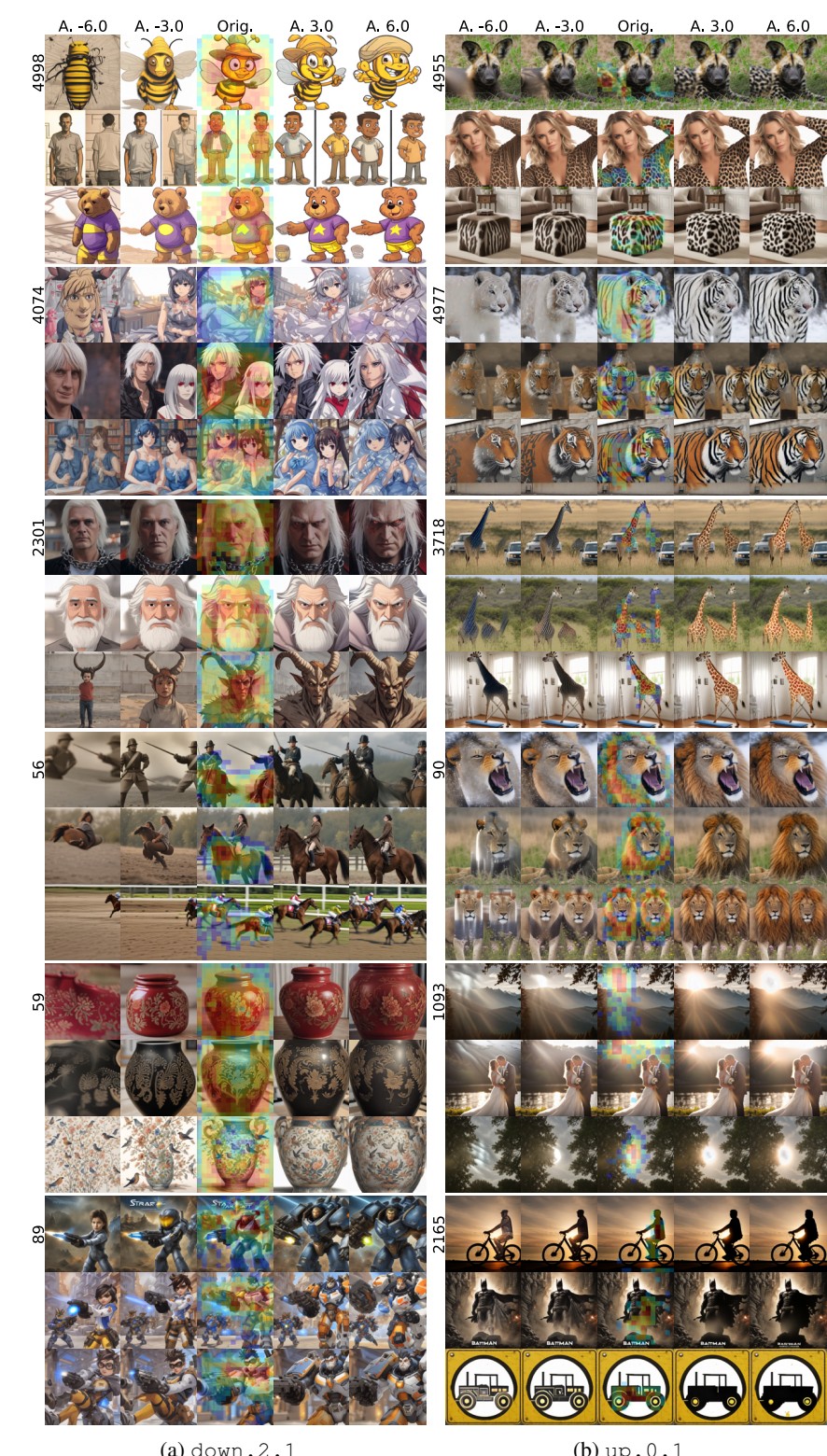

(a) down.2.1          (b) up.0.1

Figure 8: We visualize 6 features for down.2.1 (a) and up.0.1 (b). We use 5 columns for each transformer block and three rows for each feature. We visualize three samples from the top 5% quantile of activating dataset examples (middle) together a feature ablation (left) and a feature enhancement (right). Captions are in Table 5.

Table 5: Prompts for the top 5% quantile examples in Fig. 8

| Block | Feature | Prompt |
|-------|---------|--------|
| down.2.1 | 4998 | A cartoon bee wearing a hat and holding something. |
| | 4998 | Two cartoon pictures of the same man with his hands in his pockets. |
| | 4998 | A cartoon bear with a purple shirt and yellow shorts. |
| | 4074 | An anime character with cat ears and a dress. |
| | 4074 | Two anime characters, one with white hair and the other with red eyes. |
| | 4074 | An anime book with two women in blue dresses. |
| | 2301 | A man with white hair and red eyes holding a chain. |
| | 2301 | An animated man with white hair and a beard. |
| | 2301 | The character is standing with horns on his head. |
| | 56 | Two men in uniforms riding horses with swords. |
| | 56 | A woman riding on the back of a brown horse. |
| | 56 | Two jockeys on horses racing down the track. |
| | 59 | A red jar with floral designs on it. |
| | 59 | An old black vase with some design on it. |
| | 59 | A vase with birds and flowers on it. |
| | 89 | StarCraft 2 is coming to the Nintendo Wii. |
| | 89 | Overwatch is coming to Xbox and PS3. |
| | 89 | The hero in Overwatch is holding his weapon. |
| up.0.1 | 4955 | An African wild dog laying in the grass. |
| | 4955 | The woman is posing for a photo in her leopard print top. |
| | 4955 | An animal print cube ottoman with brown and white fur. |
| | 4977 | A white tiger with blue eyes standing in the snow. |
| | 4977 | A bottle and tiger are shown next to each other. |
| | 4977 | A mural on the side of a building with a tiger. |
| | 3718 | Giraffes are standing in the grass near a vehicle. |
| | 3718 | Two giraffes standing next to each other in the grass. |
| | 3718 | A giraffe standing next to an ironing board. |
| | 90 | A lion is roaring its teeth in the snow. |
| | 90 | A lion sitting in the grass looking off into the distance. |
| | 90 | Two lions with flowers on their backs. |
| | 1093 | The sun is shining over mountains and trees. |
| | 1093 | Bride and groom in front of a lake with sun flare. |
| | 1093 | The milky sun is shining brightly over the trees. |
| | 2165 | The silhouette of a person riding a bike at sunset. |
| | 2165 | The Dark Knight rises from his cave in Batman's poster. |
| | 2165 | A yellow sign with black design depicting a tractor. |

Table 6: down.2.1 first 10 and last 10 feature captions.

| Block | Feature | Caption |
|-------|---------|---------|
| down.2.1 | 0 | Organizational/storage items for documents and office supplies |
| | 1 | Luxury kitchen interiors and designs |
| | 2 | Architectural Landmarks and Monumental Buildings |
| | 3 | Upper body clothing and attire |
| | 4 | Rustic or Natural Wooden Textures or Surfaces |
| | 5 | Intricately designed and ornamental brooches |
| | 6 | Technical diagrams and instructional content |
| | 7 | Feature predominantly activated by visual representations of dresses |
| | 8 | Home decor textiles focusing on cushions and pillows |
| | 9 | Eyewear: glasses and sunglasses |
| | 5110 | Concept of containment or organized enclosure |
| | 5111 | Groups of people in collective settings |
| | 5112 | Modern minimalist interior design |
| | 5113 | Indoor plants and greenery |
| | 5114 | Feature sensitivity focused on sneakers |
| | 5115 | Handling or manipulating various objects |
| | 5116 | Athletic outerwear, particularly zippered sporty jackets |
| | 5117 | Spectator Seating in Sporting Venues |
| | 5118 | Textiles and clothing materials, focus on textures and folds |
| | 5119 | Yarn and Knitting Textiles |

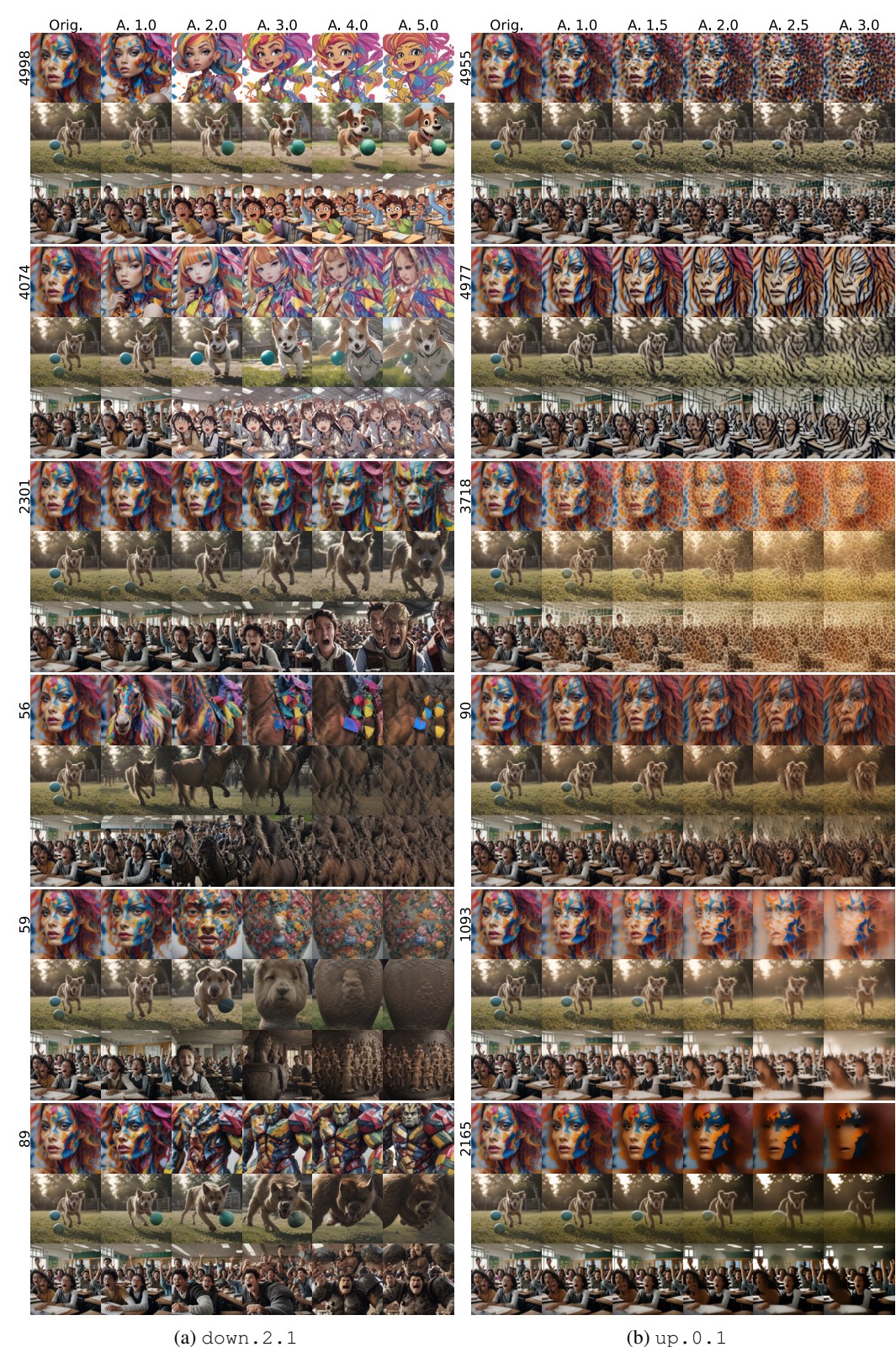

(a) `down.2.1`    (b) `up.0.1`

Figure 9: We turn on the features from Fig. 8 on three unrelated prompts "a photo of a colorful model", "a cinematic shot of a dog playing with a ball", and "a cinematic shot of a classroom with excited students".

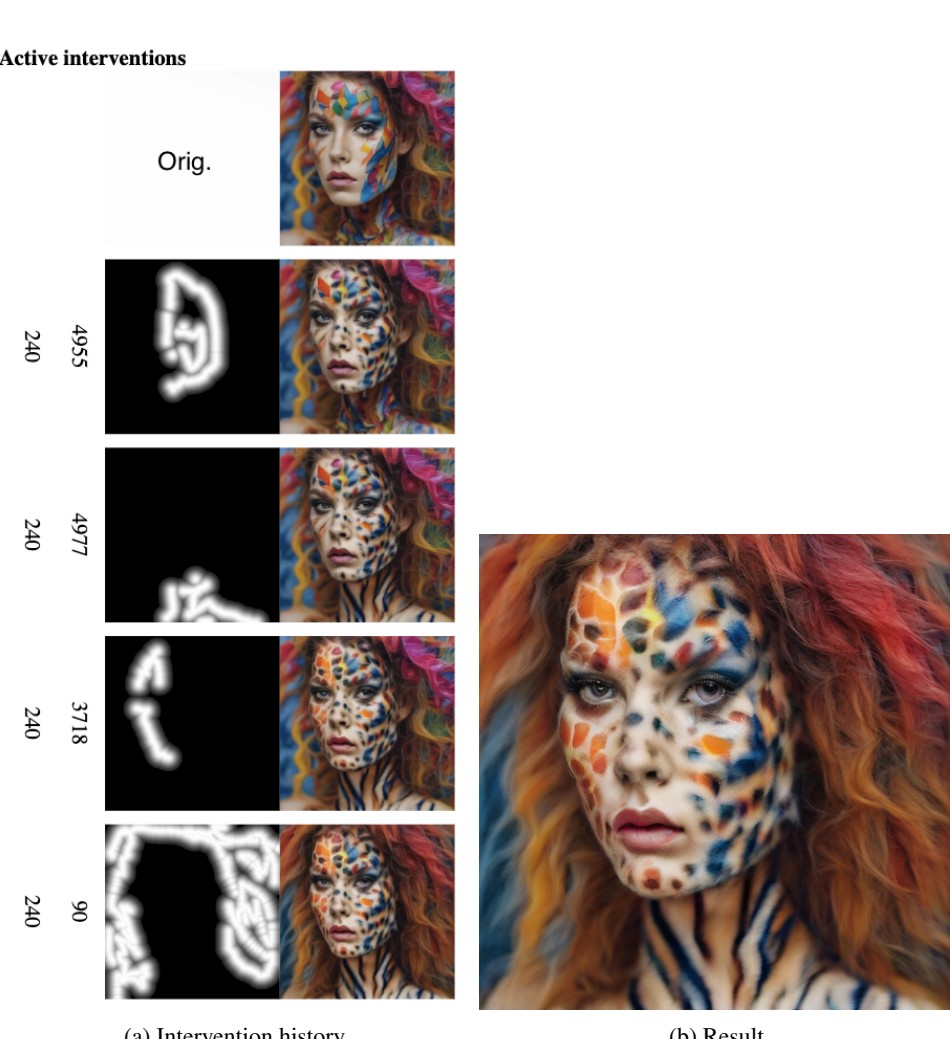

(a) Intervention history          (b) Result

Figure 10: Local edits using our research tool showcasing `up.0.1`'s ability to locally change textures in the image without affecting the remaining image. Multiple consecutive interventions are possible (a). The first in (a) row depicts the original image and each subsequent row we add an intervention by drawing a heatmap with a brush tool and then turning on the feature labelling the row only on that area. The other number (240) is the absolute feature strength of the edit. Figure (b) shows the final result in full resolution (512x512).

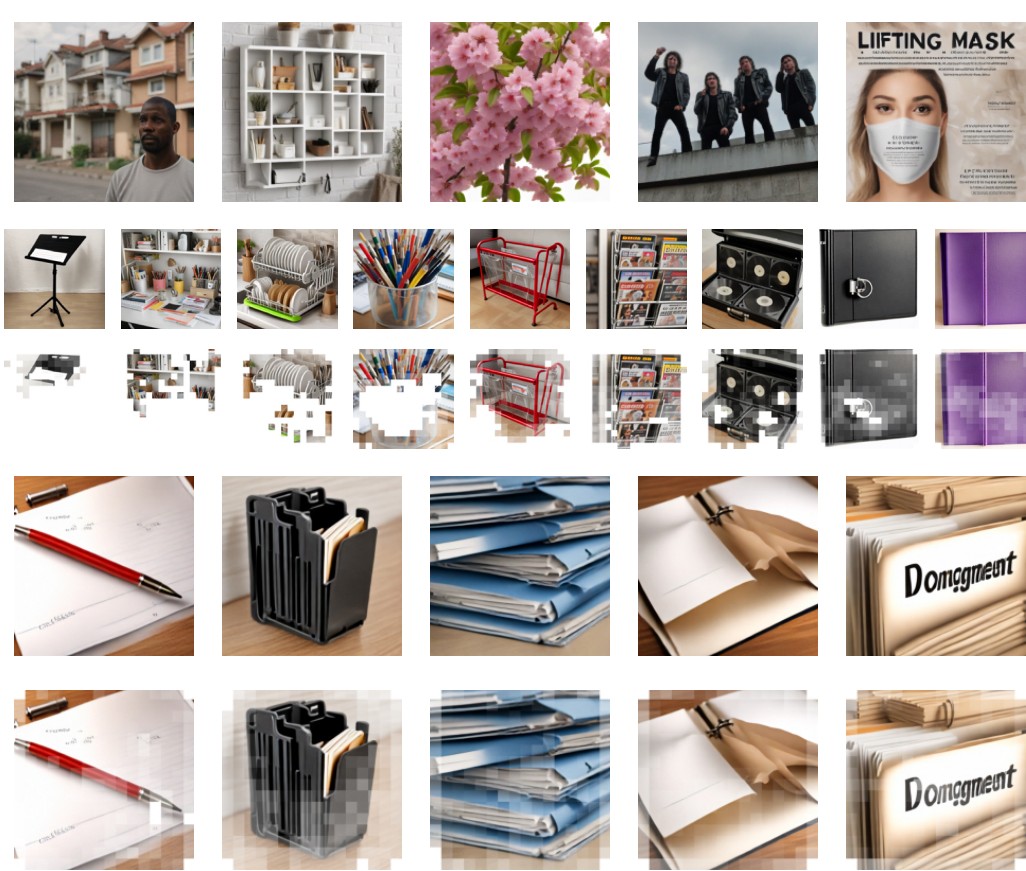

Figure 11: The images used by GPT-4o to generate captions for feature 0. From top to bottom: irrelevant images to feature 0; image progression from left to right, showing increasing activation of SAE feature 0, with low activation on the left and high activation on the right; "Coldmaps" representing the image progression; images corresponding to the highest activation of feature 0; "Coldmaps" corresponding to these highest activation images.

