# OpenReview forum: "Unpacking SDXL Turbo: Interpreting Text-to-Image Models with Sparse Autoencoders"
_ICLR.cc/2025/Conference — ICLR 2025 Conference Withdrawn Submission_

### Official Review · Reviewer_UWXd · 2024-10-27

**Soundness:** 3
**Presentation:** 1
**Contribution:** 2
**Rating:** 3
**Confidence:** 3

**Summary:**

The paper investigates the possibility of explainability and interpretability of text-to-image diffusion models using sparse autoencoders.
To this end.
The work focuses on SD-XL as their subject for analysis. Through extensive experiments the authors aim to deduct conclusion about the inference process in the network.

**Strengths:**

* The paper has extensive section on their methdology and related work
* The induction is well written and highlight the relevance of this topic
* There is clear novelty in the paper as it tries to transplant methods for interpreting generative models into the text to image sphere.

**Weaknesses:**

* The writing of this paper has a big problem, as it feels like a journal-sized paper being squeezed forcefully into the conference's page limit.
* The naming of the layers is confusing, while it can somewhat be deducted what  things like "up.0.1" mean, it becomes remarkably unclear since Figure 2 is on page 19 of the paper.
* The same disorientation during reading happens with the entire results section. As the paper is heavily based on interpreting visual heat maps, putting virtually all visual aids and result tables into the appendix is structurally not-ideal to put it mildly. It also makes the paper without the supplementary material virtually unreadable, which really bends the rules of the page limit in unintended ways. I would strongly suggest distilling the math part down, referencing the appendix for details and adding the illustrations into the main paper, to cleanly separate optional, but important information (Appendix) and critical information in the main text.
* The results are not obvious from tables, and the fact that focus was directed to a few layers makes me concerned that these are very noisy and this not really insightful.

* Minor point:
Some citation like [Pernias et. al 2023] are arxiv preprints of peer reviewed papers, it would be good to include the pper reviewed bibtex instead of arxiv whenever possible.

**Questions:**

* Did you do any ablation on the number of steps, on how it changes the behavior of the layers?
* From experimentation with diffusion models, I have learned that the time step has a great impact on the behavior of the model, in terms of which features are processed when. Hence, I wonder how the analysis of specific time-steps would change your interpretations, do you have any hypothesis?

---

### Official Review · Reviewer_7uHh · 2024-11-01

**Soundness:** 2
**Presentation:** 2
**Contribution:** 2
**Rating:** 3
**Confidence:** 3

**Summary:**

This research paper investigates the use of sparse autoencoders (SAEs) to interpret the inner workings of text-to-image diffusion models, specifically focusing on SDXL Turbo. The authors demonstrate that SAEs can effectively extract interpretable features from the model's intermediate representations, revealing causal relationships between these features and the generated images. They identify distinct roles for different transformer blocks within the model's architecture, with some blocks specializing in image composition, detail addition, and style. The study contributes to a better understanding of the internal mechanisms of diffusion models, potentially leading to improved control and manipulation of image generation.

**Strengths:**

1. The authors investigated the possibility of using sparse autoencoders (SAEs) to learn interpretable features for a few-step text-to-image diffusion model, SDXL Turbo. This research is pretty novel to me and also important for the development of T2I diffusion models.

2. The authors created a library called SDLens that allows users to cache and manipulate intermediate results of SDXL Turbo's forward pass. Along with this, they also developed visualization techniques to analyze the interpretability and causal effects of learned features.

3. They created an automatic feature annotation pipeline for the transformer block that appeared responsible for image composition. Based on these, they performed a quantitative analysis of the SAE's learned features.

**Weaknesses:**

1. The paper relies heavily on qualitative analysis, particularly through the visual inspection of generated images. The texture score and color activations are not well-defined in this paper.

2. Although this research is interesting, it is only evaluated the SDXL-Turbo model. Somehow, it's limiting the generalizability of this paper. Readers would wonder whether the same findings can be verified over the conventional SD (1.x, 2.x, 3.x, XL) models. Even for the few-step models, there are also quite lots of choices, including LCM[1], TCM[2], SwiftBrush[3], Diff2GAN[4], etc.

3. Actually, similar observations have been explored in previous papers on SD 1.4/1.5 models, where they also try to identify the functioning of each layer or block. I would like to see the connections with these previous works[5,6,7] and how is this paper different from them. Also, the authors are encouraged to include more applications for these new findings such as applying them to the T2I generation, text-based image editing, T2I personalization, etc. That will make the view of this paper much broader and help the readers to develop new techniques based on your findings.

[1] Latent Consistency Models Synthesizing High-Resolution Images with Few-step Inference
[2] Truncated Consistency Models
[3] swiftbrush: one-step text-to-image diffusion model with variational score distillation
[4] Diffusion2GAN: Distilling Diffusion Models into Conditional GANs
[5] P+: Extended Textual Conditioning in Text-to-Image Generation
[6] ProSpect: Prompt Spectrum for Attribute-Aware Personalization of Diffusion Models
[7] An Image is Worth Multiple Words: Multi-attribute Inversion for Constrained Text-to-Image Synthesis

**Questions:**

Please refer to the weaknesses. I mainly concern about the limitation of this paper, where it's only focusing on the usage of SDXL-turbo model. Also the evaluation metrics are not well-defined and not convincing. Furthermore, the similar findings are shown in previous papers, which is not super surprising for me. It seems this paper mainly introduce the SAE tool to the readers.

---

### Official Review · Reviewer_qt7A · 2024-11-03

**Soundness:** 3
**Presentation:** 3
**Contribution:** 3
**Rating:** 6
**Confidence:** 4

**Summary:**

The authors made an interesting attempt to understand the inner features of the SDXL Turbo using Sparse auto-encoders (SAEs). They have managed to show that different blocks within the model have specialized functions. Specifically, they found that the down.2.1 block is deals with image composition, the up.0.1 block deals with color, illumination, and style, and the up.0.0 block deals with adding local details. This work is indeed a significant first step in unraveling the internals of generative text-to-image models and highlights the potential of SAEs in the visual domain.

**Strengths:**

1. Good presentation! The inclusion of sufficient information in Appendix, supplementary materials, and etc. is commendable as it helps to support the claims made in the main paper.

2. While the analysis of using SAEs to understand the Unet of SDXL Turbo may not be comprehensive. It is still the first effort to interpret the intermediate representations of UNet. And the result is pretty interesting and open up new avenues for research in this area.

**Weaknesses:**

1. Some typos should be fixed. i.e., Figure 14 SDLX ->SDXL.

2. Some conclusions of the work is similar to [1]. To make their contribution more distinct, the authors could provide more intuitive examples of their observations and potential applications.

3. The experiments conducted appear to be somewhat limited. While focusing on down.2.1, up.0.1, up.0.0, and mid.0 is a good start, it leaves the question of what is happening in the other layers unanswered. A more comprehensive analysis of all the layers would provide a more complete understanding of the model's internal mechanics.


[1] P+: Extended Textual Conditioning in Text-to-Image Generation， https://arxiv.org/abs/2303.09522

**Questions:**

1. Why choose 1.5M prompts from LAION-COCO instead of other scales or prompt sources?

2. Why choose SDXL Turbo instead of SD1.5?

---

### Official Review · Reviewer_59de · 2024-11-04

**Soundness:** 2
**Presentation:** 2
**Contribution:** 2
**Rating:** 5
**Confidence:** 4

**Summary:**

This work has proposed a interpretability method in a diffusion model for text-to-image synthesis, SDXL-Turbo,  by using sparse autoencoder originally developed for language models, which allow inspection of the intermediate results of the forward pass. And, It shows several interpretations on each transformer block at different feature stages.

**Strengths:**

It provides several interpretations in the diffusion model, SDXL-Turbo.

**Weaknesses:**

1. How do we ensure whether those figures are cherry-picked or not? It would be better to provide a real-time demo program in the colab to attempt the visualization.

2. Whenever performing feature interpretability, it is not convenient to train the SAE for the specific models. We can observe the role of each transformer at different locations (i.e., feature-level) by just visualizing the intermediate feature maps. I doubt the usefulness and practicality of the proposed interpretability in the community. It would be better for authors to present the use cases.

3. **Generality**:

This work leverages and investigates only transformer blocks in the U-Net of SDXL-Turbo, which has convolution layers as well. I wonder why they investigate only trasnformers, excluding convolutinal features. I guess the transformer condition on text prompt. Then, this work is solely for the text conditioning. Furthermore, SDXL-Turbo is not representaive diffusion model, thus this work is only limited under the SDXL-Turbo.. It would be better to investigate whether the proposed interpretability methods can be applied to diffusion transformer methods such as Pixart-alpha/sigma, SD3, and Flux models.

**Questions:**

What is means the number in the leftmost of Fig.1? Does this mean class ID?

---

### Official Review · Reviewer_gbv4 · 2024-11-05

**Soundness:** 3
**Presentation:** 3
**Contribution:** 4
**Rating:** 5
**Confidence:** 3

**Summary:**

This paper investigates the image generation processes of SDXL-Turbo. Inspired by recent advancements in large language models, this paper adopts the concept of sparse autoencoders to analyze the specific roles of individual network layers. Building on this high-level approach, they design several qualitative and quantitative metrics for detailed analysis. As a result, the paper identifies layers specialized for constructing image composition, local details, color, style, and other attributes.

**Strengths:**

1. The use of sparse autoencoders is thoughtfully adapted for application in text-to-image diffusion models.

2. This paper introduces various investigative techniques using sparse autoencoders, allowing for in-depth analysis and discussion.

**Weaknesses:**

1. Limited Adaptability of the Proposed Method

The primary concern is that this paper only examines the generation process of SDXL-Turbo in a single diffusion step. While SDXL-Turbo is indeed a prominent text-to-image diffusion model based on a U-Net architecture with single-step generation capabilities, the trend in diffusion models has shifted towards transformer-based architectures. Recently developed models like the Pixart series [A,B], SDM3 [C], and Flux [D] exemplify this shift. Additionally, many models now also support multi-step generation options (e.g., one-step, four-step, eight-step), making single-step generation a less typical scenario. Given these trends, the scope of this paper may be too narrow to draw broad conclusions applicable to a wide range of diffusion models. Expanding the study to include transformer-based diffusion models and multi-step generation scenarios would enhance the generalizability and impact of this work.

2. Scale of Experiments

While the paper offers extensive qualitative and quantitative analysis, the experiments seem to rely on a small number of samples. This raises concerns about the robustness of the conclusions, as insights drawn from limited data may not be generalizable.

3. Lack of Application for Insights

The study reveals that certain layers are specialized for tasks such as image composition and style. However, it would be beneficial to demonstrate the practical value of these findings. For instance, insights into layer specialization could be applied to improve generation efficiency by pruning less essential layers or enhancing image quality by focusing on layers responsible for composition or style. While this point is less critical than the previous two, as presenting a valuable investigation method itself has large academic value.

References

[A] PIXART-α: Fast Training of Diffusion Transformer for Photorealistic Text-to-Image Synthesis, ICLR 2024

[B] PIXART-Σ: Weak-to-Strong Training of Diffusion Transformer for 4K Text-to-Image Generation, ECCV 2024

[C] Scaling Rectified Flow Transformers for High-Resolution Image Synthesis (SDM3)

[D] https://blackforestlabs.ai/announcing-black-forest-labs/

**Questions:**

None

---

### Note · Authors · 2024-11-15

I have read and agree with the venue's withdrawal policy on behalf of myself and my co-authors.